# MoodAngels: A Retrieval-augmented Multi-agent Framework for Psychiatry Diagnosis

**Mengxi Xiao**[1,2]**, Ben Liu**[3]**, He Li**[3]**, Jimin Huang**[5]**, Qianqian Xie**[1,2]**,**
**Xiaofen Zong**[4*]**, Mang Ye**[3,6*]**, Min Peng**[1,2*]
[1]School of Artificial Intelligence, Wuhan University
[2] Center for Language and Information Research, Wuhan University
[3]School of Computer Science, Wuhan University
[4]Department of Psychiatry, Renmin Hospital of Wuhan University
[5]The Fin AI
[6]Taikang Center for Life and Medical Sciences, Wuhan University

## Abstract

The application of AI in psychiatric diagnosis faces significant challenges, including the subjective nature of mental health assessments, symptom overlap across disorders, and privacy constraints limiting data availability. To address these issues, we present MoodAngels, the first specialized multi-agent framework for mood disorder diagnosis. Our approach combines granular-scale analysis of clinical assessments with a structured verification process, enabling more accurate interpretation of complex psychiatric data. Complementing this framework, we introduce MoodSyn, an open-source dataset of 1,173 synthetic psychiatric cases that preserves clinical validity while ensuring patient privacy. Experimental results demonstrate that MoodAngels outperforms conventional methods, with our baseline agent achieving 12.3% higher accuracy than GPT-4o on real-world cases, and our full multi-agent system delivering further improvements. Evaluation in the MoodSyn dataset demonstrates exceptional fidelity, accurately reproducing both the core statistical patterns and complex relationships present in the original data while maintaining strong utility for machine learning applications. Together, these contributions provide both an advanced diagnostic tool and a critical research resource for computational psychiatry, bridging important gaps in AI-assisted mental health assessment.

## 1 Introduction

Mental diseases [1], with their high prevalence and profound societal impact, pose a major public health challenge by severely impairing quality of life. Accurate diagnosis [2] is essential for timely intervention and effective treatment [3], yet the complexity and variability of symptoms make it particularly difficult [4], highlighting the need for advanced diagnostic tools to aid clinicians. Among mental diseases, mood disorder, including conditions like depression and bipolar disorder, is critical due to its high prevalence and the significant overlap of symptoms with other psychiatric conditions [5]. Correctly diagnosing mood disorders is crucial, as it influences the diagnostic process for other disorders [6]; for example, symptoms like difficulty concentrating may signal neurodevelopmental disorders only if they occur outside of depressive episodes. Given their prevalence and severe consequences, including suicide risk and chronic disability, mood disorders represent a significant burden on individuals and healthcare systems.

---

*Correspondence to: Min Peng <pengm@whu.edu.cn>, Mang Ye <yemang@whu.edu.cn>, Xiaofen Zong <zongxiaofen@whu.edu.cn>

39th Conference on Neural Information Processing Systems (NeurIPS 2025).

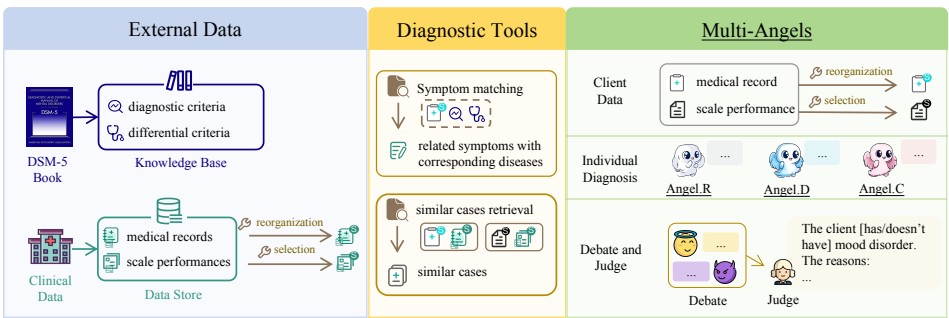

Figure 1: The MoodAngels framework. Diagnostic agents include Angel.R, Angel.D, Angel.C, and multi-Angels.

While large language models (LLMs) and LLM-based agents demonstrate strong capabilities in medical domains through robust textual analysis and decision-making, their application to psychiatry faces unique challenges. General medical diagnostic agents typically rely on concrete medical records or biological test results [7, 8], resources largely unavailable in psychiatric practice. These limitations interact with the field's inherent uncertainties, including significant symptom overlap across disorders [9–11] and the absence of definitive biomarkers [12, 13] that are standard in other medical specialties, creating fundamental barriers for AI implementation. These diagnostic difficulties are intensified by psychiatry's reliance on nuanced interpretation of subjective clinical data, which contrasts sharply with the structured evidence typically used to train LLM-based agents [7, 8]. Moreover, the situation faces additional complications from data accessibility constraints. Clinical diagnostic information, while rich in potential insights, contains sensitive patient data that cannot be publicly shared, creating a critical bottleneck for AI-driven psychiatric research. Taken together, this dual challenge of diagnostic complexity and data scarcity underscores the urgent need for specialized systems capable of emulating clinicians' probabilistic reasoning under conditions of imperfect information [14].

To address these challenges, we propose **MoodAngels** (Figure 1), the first retrieval-augmented multi-agent framework for mood disorder diagnosis, which enhances the diagnostic process through granular-scale analysis and multi-step verification. The system tackles the inherent challenges of psychiatric diagnosis, reliance on potentially unreliable self-reported and clinician-estimated scales (with optional medical records), by decomposing traditional scale scoring into item-level analysis. Using Pearson correlation, we identify the top 5% most statistically significant questions for mood disorders, categorizing them into five diagnostic groups (depression, suicidal ideation, energy/interest loss, anxiety, and insomnia). By analyzing consistency within these groups, MoodAngels can resolve diagnostic discrepancies (e.g., when self-reported depressive symptoms conflict with clinical observations) through additional behavioral marker validation.

To address symptom overlap, we structure DSM-5 diagnostic criteria [15] into a retrievable knowledge base, augmented with anonymized clinical data to incorporate expert judgment and handle ambiguous cases. We develop three diagnostic variants to balance historical reliance with individual variability: Angel.R (no reference), Angel.D (case display), and Angel.C (comparative analysis). Our final multi-Angels model synthesizes their independent diagnoses through debate, combining computational efficiency with clinical nuance.

Beyond diagnostic frameworks, clinical data scarcity presents another research challenge. Despite longstanding interest in depression and bipolar detection, existing methods predominantly analyze social media posts [16–18], which may be distorted by social norms or exaggeration. To enable accurate AI-driven psychiatric diagnosis and early detection, we construct the open-source synthetic dataset **MoodSyn**[2] with 1,173 synthetic psychiatric cases, containing: selected five groups of top-related scale data, total scores from 13 common mental disorder scales, and mood disorder labels. Through a comprehensive evaluation of quality, ML efficiency, and privacy protection, we demonstrate that this synthetic dataset maintains high fidelity while safeguarding visitor confidentiality.

---

[2]Code and synthetic data sample are available in MoodAngels.

We evaluate the effectiveness of our framework on 561 real-world clinical cases and 140 synthetic cases. Experimental results on both real cases and synthetic data demonstrate the superiority of our diagnostic process, with all versions of our agents outperforming bare LLMs that rely solely on client performances as context. Notably, on real data, even our raw agent, Angel.R, achieves a 12.3% higher diagnostic accuracy than our backbone LLM, GPT-4o. The multi-agent framework further enhances performance, surpassing the accuracy of all single-agent variants and showing considerable improvement on hard cases. Ablation studies confirm the individual contributions of our medical record analysis and scale selection processes, highlighting their critical roles in improving diagnostic precision.

Our contributions are summarized as follows:

- We propose the first psychiatry diagnosis agent framework, MoodAngels, specifically targeting mood disorders. Our agents effectively address challenges such as the lack of objective diagnostic tools, symptom overlaps, and clients misjudging or misrepresenting their mental states, achieving high diagnostic accuracy.

- We present MoodSyn, an open-source dataset of 1,173 synthetic psychiatric cases that provides clinically plausible alternatives to sensitive patient data while preserving key statistical patterns for mood disorder detection research.

- Experimental results validate the effectiveness of MoodAngels, with all agent variants outperforming bare LLMs. Notably, on real-world data, even our raw agent surpasses GPT-4o by 12.3% in accuracy, while the multi-agent framework further enhances performance. Ablation studies highlight the contributions of the granular-scale analysis and multi-step verification, underscoring their importance in achieving robust diagnostics.

## 2 Retrieval-augmented Multi-agent Framework

In this section, we propose a retrieval-augmented multi-agent framework tailored for the diagnosis of mood disorders[3], integrating structured clinical knowledge with dynamic decision-making. The framework consists of three specialized diagnostic agents, each performing an independent diagnostic process based on reorganized medical records and selected diagnostic scales. These agents employ two core analytical tools: a Granular-Scale Analysis module that matches symptoms to diagnostic criteria at a fine-grained level, and a Case-Based Retrieval module that retrieves relevant prior cases from a curated knowledge base. The diagnostic opinions of the agents are then synthesized through a structured debate mechanism to form a final judgment.

The key innovations of MoodAngels focus on the depth of symptom analysis and the design of diversified diagnostic reasoning processes. Unlike conventional methods that rely solely on total scale scores, the Granular-Scale Analysis method captures intra-scale symptom patterns, which supports a more personalized and accurate understanding of individual differences. This aspect is particularly important in psychiatric assessment, where such variability is often overlooked. In addition, although case retrieval is commonly applied in general medical diagnosis, psychiatric evaluation requires more cautious application due to its subjectivity and high inter-individual variability. To address this challenge, the three diagnostic agents are designed with different levels of reliance on historical cases. This ensures that prior knowledge contributes meaningfully to the diagnostic process without dominating it. The complementary perspectives of the agents are ultimately integrated through structured debate, improving both the robustness and interpretability of the final diagnostic outcome.

### 2.1 Granular-scale Analysis

The primary challenge in constructing the diagnostic framework involves accurately identifying information that reliably reflects the visitor's actual condition. This difficulty arises because psychiatric diagnoses depend on self-reported data and clinician-estimated scales (supplemented by optional medical records), which carry inherent risks of misjudgment or misrepresentation.

To address this challenge, we introduce an innovative granular-scale analysis approach[4]. Rather than evaluating scales solely through total scores, we decompose them into individual item-level

---

[3]The problem definition is provided in Appendix B.1.

[4]We provide simple descriptions of scales and the selection process of relevant questions in Appendix B.2.

responses. We employed a comprehensive set of clinical scales to assess various aspects of mental health, including eight self-reported scales and five clinician-evaluated scales. These tools provide valuable insights into both subjective experiences and symptoms, as well as clinical observations and structured evaluations of clients' mental states and behaviors. All scales are routinely used in hospital settings for clinical diagnosis, ensuring the reliability and validity of the collected data.

To identify the most relevant questions for mood disorder diagnosis, we computed the Pearson correlation between each question's score (and total score) and the presence of a mood disorder, selecting the top 5% with the highest correlations. These questions naturally clustered into key symptom groups: depressive mood, loss of interest, anxiety, insomnia, and suicidal tendencies. These groups enhance diagnostic robustness through cross-validation and comprehensive symptom coverage. To further refine our framework, we included clinically significant PHQ-9 questions, such as phq9_Q2 (depressed mood) and phq9_Q1 (loss of interest), even if their correlation scores were slightly below the threshold, ensuring a nuanced and reliable diagnostic process.

By evaluating response consistency within each symptom group, we derive more accurate inferences about the visitor's probable condition. For instance, when a visitor reports frequent depressive symptoms on a self-assessment scale but clinicians observe no corresponding depressive signs, this discrepancy directs MoodAngels to investigate additional behavioral markers for diagnostic validation.

## 2.2 Retrieval Datastore

Since overlapping symptoms may correspond to multiple disorders, we extract and structure diagnostic and differential criteria from the *Diagnostic and Statistical Manual of Mental Disorders: DSM-5* [15], a widely recognized authority in psychiatry, to build a retrievable knowledge base. The knowledge base construction process is detailed in Appendix B.3.2.

To prevent MoodAngels from making arbitrary decisions based solely on symptom presentation, we also incorporate clinicians' diagnostic expertise by including anonymized clinical data for retrieval. These experiences are also beneficial when a visitor's symptoms are ambiguous, for historical diagnostic precedents may offer additional interpretive insights. The clinical data used in our study consists of anonymized real-world hospital cases, totaling 2804 entries. We partitioned the dataset such that 80% of the cases are used as historical cases for retrieval, while the remaining 20% serve as the test set. All clients in the dataset have completed scale assessments, although clients without diagnosed conditions do not have medical records available, and our agents are not pre-informed about this distinction. The dataset statistics are summarized in Table 1.

Table 1: Dataset statistics of real-world clinical cases.

|                    | Normal | Mood Disorder | Other Diseases | Total |
|--------------------|--------|---------------|----------------|-------|
| cases for retrieval | 1259   | 759           | 225            | 2243  |
| cases for test     | 315    | 190           | 56             | 561   |

## 2.3 Diagnostic Agents

To mitigate overreliance on past cases (which could overlook individual variability in psychiatric diagnoses), we develop three diagnostic variants with differing levels of historical dependence: Angel.R (no reference to previous cases), Angel.D (displays retrieved cases as context), and Angel.C (compares each retrieved case with the current query and returns an analysis as context). Prompts of this angels are provided in Appendix C.1. By aggregating independent diagnoses from these three agents and facilitating debate among their conclusions, our final diagnosis model, multi-Angels, bridges the gap between computational decision-making and the nuanced understanding essential for accurate psychiatric evaluations. The following parts introduce the main components of our agents:

**Symptom Matching.** To align client symptoms in medical records with DSM-5 diagnostic criteria, we process records and compute relevance between records and criteria using dense vector encoding [5]. The BGE-M3 embedder [19] is employed for its strong semantic embedding capabilities. We retrieve

---
[5]The processing steps of medical records are detailed in Appendix C.2.1, and the formulation of symptom matching is provided in Appendix C.2.2.

the top-5 most similar criteria, returning their text, classification, and similarity scores. The tool does not diagnose but provides results for agent analysis, ensuring decisions integrate quantitative data and clinical expertise, mitigating over-reliance on single metrics.

For cases with overlapping symptoms, an additional instruction prompts the agent to consider differential diagnosis, guiding systematic evaluation of potential conditions. This enhances the agent's ability to distinguish between mood disorders and other diseases.

**Scale Performance Analysis.** We diagnose the presence of mood disorder using 16 key questions selected in Section 2.1 as the most mood-relevant items. Client performances are converted from numeric scores to textual descriptions based on question content and options. For agent interpretability, performances are reorganized into coherent descriptive paragraphs, enhancing analysis effectiveness (examples are provided in Appendix C.3).

**Similar Cases Retrieval.** To leverage clinical experience from similar cases, we develop two optional tools for retrieving medical records and scales with similar performance. After performing similarity retrieval, our tools generate different outputs tailored to the type of diagnosis agent in use. For Angel.R, this tool is intentionally excluded to minimize potential interference from the diagnostic outcomes of other cases. For Angel.D, the tool directly returns the retrieved cases for reference, enabling the agent to review and draw insights from them. For Angel.C, the tool conducts a detailed comparison of similarities and differences among the retrieved cases and returns an analysis text summarizing the findings. Consistent with the symptom matching tool, we employ BGE-M3 as the retriever. This approach ensures that the tool adapts to the specific needs of each diagnosis agent, enhancing the diagnostic process while maintaining flexibility and precision.

**Multi-agent Diagnosis.** To integrate insights from all Angels and improve diagnostics, Angel.R, Angel.D, and Angel.C first provide independent decisions and reasoning. A Judge Agent consolidates their inputs. If consensus is reached, the Judge outputs the diagnosis and reasoning. For disagreements, two Debate Agents are introduced: a Positive Agent, supporting a mood disorder diagnosis, and a Negative Agent, opposing it. Both Debate Agents and the Judge access symptom matching results, scale performances, relevant cases, and the Angels' diagnosis and reasoning. In each debate round, the Positive Agent speaks first, followed by the Negative Agent. After each round, the Judge evaluates the arguments and decides whether to conclude the debate. If concluded, the Judge delivers the final diagnosis and supporting reasons. More details about the judge and a complete debate example are presented in Appendix C.4.1.

# 3 MoodSyn Dataset

The **MoodSyn** dataset addresses the critical need for clinically valid yet privacy-preserving data in computational psychiatry by introducing an open-source collection of 1,173 synthetic cases, as the statistics shown in Table 2. Each case captures the essential characteristics of psychiatric assessments through 25 carefully selected features, including 16 diagnostic questions, 8 standard scale scores, and expert-verified mood disorder labels. A data example is shown in Appendix D.1.

Table 2: Descriptive statistics of the synthetic MoodSyn dataset. Positive counts represent cases with mood disorder diagnoses, while negative counts indicate an absence of mood disorders.

|  | positive amount | negative amount | Total |
| --- | --- | --- | --- |
| cases for retieval | 687 | 419 | 1106 |
| cases for test | 73 | 67 | 140 |

**Dataset Construction.** The MoodSyn dataset is constructed on an advanced synthesis pipeline built upon the TabSyn framework [20], which integrates variational autoencoders with diffusion models through several technical innovations. The architecture employs adaptive tokenization and hierarchical encoding to maintain the complex relationships between clinical features, while a conditional denoising process preserves the nuanced symptom patterns characteristic of mood disorders. During generation, dynamic loss weighting ensures the mathematical consistency between individual question responses and their corresponding scale scores, and context-aware sampling maintains clinically meaningful feature correlations. The construction process is detailed in Appendix D.1.

**Dataset Evaluation.** Following Zhang et al. [20], we conduct a comprehensive evaluation of MoodSyn across five critical dimensions: statistical density, data quality, machine learning efficacy, privacy preservation, and logistic detectability (detailed in Appendix D.2). The results demonstrate exceptional fidelity to real clinical data, with MoodSyn accurately reproducing both core statistical patterns and complex symptom relationships while maintaining strong utility for machine learning applications. Specifically, our evaluation reveals high-density preservation of univariate and multivariate distributions matching the original data's feature patterns, robust performance in downstream diagnostic prediction tasks comparable to real data, and near-indistinguishability in rigorous detection tests. Furthermore, MoodSyn provides stronger privacy guarantees than traditional anonymization approaches through its synthetic generation process. This unique combination of statistical fidelity, clinical validity, and privacy protection establishes MoodSyn as both a reliable analytical surrogate and a valuable resource for advancing AI applications in mental health research.

# 4 Experiment

## 4.1 Experimental Settings

**Datasets.** We assess the effectiveness of our agent framework on the test set described in Table 1, which comprises 561 cases. These include 315 normal cases, 56 cases of mood disorders, and 190 cases of other mental disorders. Cases without mental disorders contain only scale data and no medical records, whereas the remaining cases include both medical records and scale data. All cases are derived from real clinical data collected from our corporate hospital, ensuring the evaluation reflects practical diagnostic scenarios.

**Baselines.** To evaluate the effectiveness of our agent, we compare it against five baseline LLMs: LLaMA3-8B-Instruct [21], Mistral-7B-Instruct-v0.3 [22], GPT-4o (gpt-4o-2024-08-06) [23], DeepSeek-V3 [24], and medgemma-27b-text-it [25]. Each LLM is provided with a standardized input consisting of: (1) A combined medical record, presented as a unified narrative. (2) A summary of scale test results, listing the client's scores across multiple psychological scales along with their clinical implications. The LLMs are prompted to determine whether the client has a mood disorder (including depression or bipolar disorder) and to provide a structured explanation in JSON format. An example of the prompt used for baseline models is shown in Figure 8.

**Hyperparameters.** The retrieval number $k$ for Angel.D and Angel.C is set to $k = 5$ by default. For each model, we employ default parameter settings, utilizing official models for open-source LLMs obtained from Hugging Face or the API from the official website. These testing procedures take place on a computational infrastructure consisting of two NVIDIA A800 Tensor Core GPUs, equipped with 80GB of memory.

**Evaluation Metrics.** Following previous methods [16], we utilize Recall, Accuracy (ACC), Matthews Correlation Coefficient (MCC) [26], and Macro F1 to evaluate the performance of various versions of MoodAngels and baseline LLMs on the mood disorder diagnosis.

More details about baseline prompts and the selection of retrieval number $k$ are provided in Appendix E.1.

## 4.2 Analysis of Experimental Results

Our diagnostic framework achieves superior performance through innovative components that transform psychiatric assessment. Unlike conventional approaches limited by total score interpretation of standardized scales, MoodAngels implements granular item-level analysis that identifies and groups mood-relevant questions, overcoming the inherent information loss of aggregate scoring. This technical advancement enables more precise detection of symptom patterns that would otherwise be obscured in traditional scale processing. The framework's structured diagnostic process represents another significant innovation, replacing the direct diagnosis generation used by baseline models with a rigorous multi-step verification system. While Angel.R establishes core functionality through DSM-5 criteria referencing, our more advanced variants incorporate clinical case retrieval and inter-agent debate, a substantial departure from the direct inference approach employed by bare LLMs.

The experimental results presented in Tables 3 and 4 demonstrate the clear advantages of this approach. We further observed that the generated reasoning demonstrates high quality, as detailed

in Appendix E.2.1. All MoodAngels variants show substantial improvements over baseline LLMs, with even our foundational agent achieving 12.3% greater accuracy than GPT-4o, underscoring the effectiveness of our redesigned diagnostic process. A deeper examination of the agent comparisons reveals important insights about psychiatric diagnosis. The performance advantage of Angel.D over Angel.R confirms the clinical value of historical cases as reference points, while the slight dip observed with Angel.C serves as a caution against over-reliance on past cases. Particularly telling are those instances where only one agent succeeds in making the correct diagnosis while others fail, highlighting both the complexity of psychiatric assessment and the complementary nature of different diagnostic approaches.

We also conduct subgroup analysis on gender and age, confirming no significant bias in predictions (detailed in Appendix E.2.2). These findings collectively validate our multi-agent framework's design, which synthesizes granular symptom analysis, structured diagnostic protocols, and balanced clinical experience integration through its collaborative debate mechanism. The framework's ultimate performance superiority emerges from this sophisticated combination of innovations, each addressing specific limitations in conventional psychiatric assessment methods while working in concert to achieve more accurate and reliable diagnoses.

Table 3: Diagnosis performances of all methods on real data.

| Model | Recall | ACC | MCC | Macro F1 |
|---|---|---|---|---|
| *in-context learning* | | | | |
| LLaMA3-8B-Instruct | 0.393 | 0.506 | 0.315 | 0.492 |
| Mistral-7B-Instruct-v0.3 | 0.456 | 0.597 | 0.411 | 0.397 |
| GPT-4o | 0.631 | 0.797 | 0.639 | 0.792 |
| Deepseek-V3 | 0.703 | 0.847 | 0.716 | 0.841 |
| medgemma | 0.67 | 0.743 | 0.57 | 0.717 |
| *GPT-4o based agents* | | | | |
| Angel.R$_{GPT-4o}$ | 0.840 | 0.920 | 0.829 | 0.913 |
| Angel.D$_{GPT-4o}$ | 0.845 | 0.923 | 0.837 | 0.917 |
| Angel.C$_{GPT-4o}$ | 0.848 | 0.914 | 0.814 | 0.906 |
| multi-Angels$_{GPT-4o}$ | **0.881** | 0.925 | 0.834 | 0.917 |
| *Deepseek-V3 based agents* | | | | |
| Angel.R$_{Deepseek-V3}$ | 0.863 | **0.927** | **0.841** | **0.920** |
| Angel.D$_{Deepseek-V3}$ | 0.864 | 0.920 | 0.823 | 0.911 |
| Angel.C$_{Deepseek-V3}$ | 0.858 | 0.922 | 0.829 | 0.914 |
| multi-Angels$_{Deepseek-V3}$ | 0.866 | 0.923 | 0.832 | 0.916 |

Table 4: Diagnosis performances of all methods on synthetic data.

| Method | Recall | ACC | MCC | Macro F1 |
|---|---|---|---|---|
| *in-context learning* | | | | |
| LLaMA3-8B-Instruct | 0.557 | 0.564 | 0.221 | 0.435 |
| Mistral-7B-Instruct-v0.3 | 0.589 | 0.636 | 0.375 | 0.563 |
| Deepseek-V3 | 0.761 | 0.821 | 0.664 | 0.816 |
| medgemma | 0.535 | 0.704 | 0.525 | 0.704 |
| *Deepseek-V3 based agents* | | | | |
| Angel.R$_{Deepseek-V3}$ | 0.778 | 0.8 | 0.601 | 0.798 |
| Angel.D$_{Deepseek-V3}$ | 0.787 | 0.807 | 0.615 | 0.805 |
| Angel.C$_{Deepseek-V3}$ | 0.816 | **0.821** | **0.642** | **0.821** |
| multi-Angels$_{Deepseek-V3}$ | **0.824** | **0.821** | **0.642** | **0.821** |

## 4.3 Ablation Study

In this section, we adapt Angel.R on real data to test the efficiency of processed medical records and selected scales. Additionally, we evaluate the stability of our framework by changing parameter settings in Appendix E.3.

### 4.3.1 Medical Record Format

In this section, we investigated whether structured medical records improve the performance of the agents. Initially, we tested different formats in the symptom matching process, which extracts diagnostic criteria from the DSM-5 based on a medical record. Comparing Setting 1 and Setting 2, we observed only a slight decline in the agents' performance. Upon further analysis of the diagnostic criteria retrieved for a specific case, we found that the extracted criteria were highly relevant and similar. In most instances, the diagnostic criteria extracted under both experimental settings were identical, demonstrating the robustness and inclusivity of our chosen retriever.

However, when medical records were presented to the agent in different formats, the agent's diagnostic performance showed significant variation. Comparing Setting 2 and Setting 3, where the same record format was used in the symptom matching step but different formats were returned to the agent, we identified cases where the diagnosis was correct in Setting 2 but incorrect in Setting 3. We found that a continuous, narrative-style medical record contains more detailed descriptions, which better reflect the severity of the client's condition. In contrast, structured medical records tend to focus on objective facts, potentially overlooking nuanced symptoms. This makes it more challenging to identify patients with atypical presentations of mood disorders. For example, a patient with severe depression may not exhibit obvious signs of low mood or mania but instead feel a sense of hopeless calmness and express suicidal ideation. This "hopeless calmness" is clearly conveyed in a narrative-style medical record but may be less apparent in a structured format.

In summary, while structured medical records offer consistency and objectivity, they may lack the nuanced details present in narrative-style records, which are crucial for identifying atypical or subtle symptoms of mood disorders. This highlights the importance of balancing structured data with rich, descriptive narratives to ensure accurate and comprehensive psychiatric diagnosis.

Table 5: Diagnosis accuracy of different medical record formats in symptom matching and agent processing steps. The numbers indicate different combinations of experimental settings for ease of reference. In the experimental analysis, these settings are directly referred to as Setting 1, Setting 2 and Setting 3.

| | Symptom Matching | Return to Agent | ACC | | MCC | |
|---|---|---|---|---|---|---|
| 1 | unstructured | unstructured | 0.920 | \ | 0.829 | \ |
| 2 | structured | unstructured | 0.918 | 0.002↓ | 0.822 | 0.007↓ |
| 3 | structured | structured | 0.914 | 0.006↓ | 0.814 | 0.015↓ |

### 4.3.2 Selection of Scales

In this section, we compared the diagnostic accuracy of Angel.R when using two different input strategies: providing 16 selected questions most relevant to mood disorders from the 13 scales, and providing the total scores of all 13 scales without any filtering. The results in Table 6 reveal a significant performance drop of 6.8% in accuracy when using the unfiltered total scores, highlighting the importance of question selection. This substantial decline underscores that the inclusion of less relevant or noisy data from unrelated questions can adversely impact the agent's diagnostic precision. By focusing on the most pertinent questions, our agent achieves more reliable and accurate mood disorder assessments.

Table 6: Diagnosis accuracy of selected and unselected scales used in the agent processing steps. The numbers indicate different combinations of experimental settings for ease of reference. In the experimental analysis, these settings are directly referred to as Setting 1 and Setting 4.

| | Scale Used | ACC | | MCC | |
|---|---|---|---|---|---|
| 1 | selected | 0.920 | \ | 0.829 | \ |
| 4 | unselected | 0.852 | 0.068↓ | 0.708 | 0.121↓ |

## 4.4 Case Study

To demonstrate the robustness and diagnostic capabilities of our agents, we present two critical scenarios: inter-scale conflicts and overlapping symptoms. In inter-scale conflict cases, clients

may misjudge their condition in self-reported scales or, due to certain personality traits or stress responses, may not fully disclose their true feelings to clinicians. In such situations, our approach of grouping and analyzing similar questions across different scales proves invaluable, enabling the agent to identify inconsistencies and arrive at a more accurate diagnosis. Additionally, we examine cases with overlapping symptoms, where a single symptom may be indicative of multiple diseases, significantly complicating the diagnostic process. In these instances, simple symptom matching alone may fail to pinpoint the accurate disease. However, by leveraging historical case experiences and employing a multi-agent debate framework to explore various possibilities, MoodAngels achieves a more nuanced and precise diagnosis, bringing it closer to the underlying truth. These case studies highlight the strengths of our method in handling complex real-world diagnostic challenges. The intuitive presentations of these cases are shown in Appendix E.4.

**Inter-Scale Conflicts.** We analyze a case where self-reported scales conflict with clinician-evaluated performances. Although the client scored 11 on the PHQ-9, suggesting moderate depression, self-reports are subjective. Clinician evaluations across multiple professional scales (e.g., HAMD, HAMA, BPRS) revealed no depressive symptoms or energy decline, providing more objective diagnostic assurance. Historical cases with similar self-reported depression but no clinician-confirmed symptoms further support this conclusion. While the client reported mild to moderate anxiety, which correlates with mood disorders, it was insufficient for a diagnosis, especially given the clinician's negative findings. The absence of clinician-confirmed symptoms led our agent to conclude no mood disorder, demonstrating its ability to resolve inter-scale conflicts through comprehensive analysis of grouped questions and historical data.

**Overlapping Symptoms.** We analyze a particularly challenging diagnostic scenario with overlapping symptoms. While the client's self-reports indicated negative emotions, suicidal tendencies, and loss of interest, the clinician found none of these. However, the medical record confirmed the self-reports, also noting delusions and self-talk. Symptom matching with DSM-5 criteria further complicated the diagnosis, as the top five matched disorders spanned Mood Disorder, Personality Disorder, and Neurocognitive Disorder, none of which included the actual diagnosis of Schizophrenia. Only through historical case retrieval and multi-agent debate did the Judge Agent identify that the client had concealed symptoms during clinical interviews, ultimately leading to the correct Schizophrenia diagnosis. This case highlights the difficulty of diagnosing complex cases with overlapping symptoms and underscores our agent's strength in integrating diverse data sources to uncover hidden diagnostic patterns.

## 4.5   Error Analysis

We examined cases where the three single angels disagreed and instances where the multi-agent debate still resulted in incorrect diagnoses. These cases typically involved borderline symptoms or conflicts between medical records and scale performances. Conflicts between medical records and scale performances are particularly challenging, as they often require additional information to make a more confident diagnosis.

**Borderline Cases.** Some clients' self-reported and clinician-evaluated scale performances were highly consistent, both indicating mild degrees of depression, anxiety, insomnia, and loss of interest. After discussion with our coauthor expert, we concluded that while the client does not currently meet the diagnostic threshold for major depressive disorder, they fall on the borderline between normal and mild depression. Such cases warrant close attention and monitoring.

**Conflicts between Medical Record and Scale Performances.** In some cases, the medical record indicates severe symptoms, while both self-reported and clinician-evaluated scales show minimal or no symptoms related to mood disorders, with most relevant questions scoring zero. This discrepancy may arise when the client is in remission from a severe mood disorder, when medication temporarily suppresses negative emotions, or when the client has recovered. Recognizing this challenge, we enhanced our agents to flag such cases for clinicians, prompting further investigation. While the agents may misjudge the presence of a mood disorder in these scenarios, this additional signal ensures that critical cases are not overlooked, thereby improving diagnostic reliability and clinical utility.

# 5 Conclusion

In conclusion, we propose MoodAngels, the first specialized multi-agent framework for mood disorder diagnosis that addresses key challenges in psychiatry through granular-scale analysis and structured multi-step verification, achieving superior accuracy over existing methods. Our framework is complemented by MoodSyn, an open-source synthetic dataset of 1,173 clinically validated cases that enables research while preserving privacy. Experimental results demonstrate the effectiveness of our approach, with the baseline agent outperforming GPT-4o by 12.3% and the full multi-agent system achieving even greater accuracy. These contributions advance AI applications in mental health by providing both an effective diagnostic framework and a valuable research resource that addresses critical gaps in computational psychiatry.

## Ethical Statement

All medical record de-identification was conducted in an offline environment, ensuring no data was processed on external or networked systems. To further protect privacy, the medical records presented in this paper have undergone event and symptom obfuscation, along with partial modification. Additionally, all cases used in this study were included with the explicit consent of the clients, strictly for academic research purposes.

**Evaluators' Background.** Our evaluation team consists of coauthors who are experts in the field, led by a professional attending physician and professor with over 20 years of clinical experience. This lead evaluator is affiliated with one of the most authoritative hospitals in our country, bringing unparalleled expertise and credibility to the evaluation process. Their deep understanding of psychiatric disorders and extensive clinical background ensured a rigorous and reliable assessment of the knowledge base.

## Limitations

Our study is limited by the available data, which only includes client medical records and scale scores, restricting our ability to fully replicate the comprehensive diagnostic process employed by clinicians. Clinician diagnoses typically involve around one hour of patient interviews, during which additional clinician-evaluated scales are completed and more granular judgments are made based on real-time interactions. While our approach cannot fully capture this in-depth process, it still provides valuable insights by leveraging existing data to support mood disorder diagnosis, offering a promising tool for assisting clinicians in cases where complete diagnostic information may not be available.

## Acknowledgments

This work is partially supported by Key Project of the National Natural Science Foundation of China (U23A20316), National Key Research and Development Program of China (2024YFC3308400), and CCF-Tencent Rhino-Bird Open Research Fund (CCF-Tencent RAGR20250115). We gratefully acknowledge all clinicians who participated in the controlled reader study for their valuable contributions to model evaluation.

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

# A    Related Work

## A.1    LLM-based Depression and Bipolar-disorder Detection

Recent advancements in depression detection [27, 28] have primarily focused on leveraging LLMs to address the challenges of psychiatric diagnosis. Chen et al. [29] proposed structuring clinical interviews into a directed acyclic graph to enable automatic diagnosis, though this approach may struggle with cases where clients misrepresent their conditions. Social media platforms have also become a valuable resource for depression detection due to the abundance of user-generated content. Farruque et al. [16] fine-tuned a pre-trained language model on specific datasets to detect depressive symptoms from self-disclosed tweets, while Wang et al. [17] employed LLMs to identify depression-related text and predict depression levels from Reddit posts. Additionally, Lee et al. [18] utilized historical mood swings from users' past social media activities to detect bipolar disorder. Efforts to enhance data quality and model performance have further explored synthetic data generation; for instance, Bucur [30] used LLMs to generate synthetic data for BDI-II symptoms, enriching datasets with semantic diversity and emotional experiences unique to Reddit posts. These approaches collectively highlight the potential of LLMs and social media data in advancing depression detection, though challenges remain in ensuring accuracy and addressing client misrepresentations.

## A.2    Medical Diagnosis Agents

Recent studies have begun to explore the potential of LLMs [31–34] in medical diagnosis, though these approaches often face limitations when applied to psychiatric contexts. Li et al. [7] proposed a general disease diagnosis framework that leverages medical examination reports, such as blood tests and cell staining, to retrieve similar cases and rules. A doctor agent then evaluates the information to make a diagnosis, and correctly diagnosed cases are added to a historical database for future retrieval. However, this method is unsuitable for psychiatric diagnosis due to its lack of patient data anonymization, which could lead to privacy concerns and patient resistance. Additionally, psychiatric diagnosis lacks objective diagnostic tools and often involves overlapping symptoms, making rule-based approaches ineffective. Similarly, Wang et al. [8] enhanced agent expertise using external knowledge from the National Institutes of Health for conditions like diarrhea and bronchitis. Their agent-specialist outputs a probability distribution over possible diagnoses based on patient-reported symptoms. However, this approach struggles with psychiatric cases where clients may misrepresent or inaccurately estimate their mental state, highlighting the need for more nuanced frameworks tailored to mental health diagnoses.

# B    Details of MoodAngels Framework

## B.1    Problem Definition

Mood disorder diagnosis is a binary classification task aimed at determining whether a client has a mood disorder, such as depression or bipolar disorder, based on a given case of clinical data. The input for each case includes structured clinical information, which consists of medical records (which may be absent for first-time visitors) and scale performance data (which is always available). The output of the task is a diagnostic result, represented as a binary decision (yes or no), along with supporting reasons that justify the diagnosis. Formally, for a given case $C = (M, S)$, where $M$ represents the medical records and $S$ represents the scale performance data, the goal is to determine $f(C) = (\hat{y}, r)$. Here, $\hat{y} \in \{0, 1\}$ is the predicted diagnosis (1 indicating the presence of a mood disorder and 0 indicating its absence), and $r$ is the reasoning that supports the prediction. This task is particularly challenging due to the potential absence of medical records for some cases and the need to provide interpretable reasoning for the diagnostic decision.

Currently, our framework performs binary classification to determine whether a mood disorder is present. While we have extracted the DSM-5 differential criteria for more fine-grained distinctions, we do not attempt subcategory classification. This is because psychiatric diagnosis requires careful deliberation; in clinical practice, clinicians typically first identify a broad diagnostic direction before narrowing it down through further assessments. Similarly, our framework is designed to support this initial diagnostic stage.

### B.2 Granular-scale Analysis

### B.2.1 Scale Descriptions

Self-evaluated scales:

**(1) CTQ (Childhood Trauma Questionnaire).** A retrospective self-report inventory that measures the severity of childhood trauma across five domains: emotional, physical, and sexual abuse, as well as emotional and physical neglect. The scale content is available at CTQ.

**(2) DAS (Dysfunctional Attitude Scale).** A self-report measure that assesses maladaptive cognitive patterns and beliefs associated with depression. The scale content is available at DAS.

**(3) GAD-7 (Generalized Anxiety Disorder-7).** A self-report scale used to assess the severity of generalized anxiety disorder symptoms over the past two weeks. The scale content is available at GAD-7.

**(4) HCL-32 (Hypomania Checklist-32).** A self-report questionnaire designed to identify symptoms of hypomania, often used in the assessment of bipolar disorder. The scale content is available at HCL-32.

**(5) MDQ (Mood Disorder Questionnaire).** A self-report screening tool used to identify symptoms of bipolar spectrum disorders. The scale content is available at MDQ.

**(6) NSSI (Non-Suicidal Self-Injury Assessment).** A self-report measure designed to assess the frequency, methods, and motivations behind non-suicidal self-injurious behaviors. The scale content is available at NSSI.

**(7) PHQ-9 (Patient Health Questionnaire-9).** A self-administered tool used to screen for and measure the severity of depression based on the DSM-5 criteria. The scale content is available at PHQ-9.

**(8) SHAPS (Snaith-Hamilton Pleasure Scale).** A self-report measure designed to assess anhedonia, or the inability to experience pleasure, commonly associated with depression. The scale content is available at SHAPS.

Clinician-rated scales:

**(9) BPRS (Brief Psychiatric Rating Scale).** A clinician-administered scale that evaluates a broad range of psychiatric symptoms, including psychosis, depression, and anxiety. The scale content is available at BPRS.

**(10) HAMA (Hamilton Anxiety Rating Scale).** A clinician-administered scale used to measure the severity of anxiety symptoms based on observed and reported behaviors. The scale content is available at HAMA.

**(11) HAMD (Hamilton Depression Rating Scale).** A clinician-rated tool that assesses the severity of depressive symptoms, widely used in clinical and research settings. The scale content is available at HAMD-24. In this study, we employ the Chinese version of the HAMD-24, which features a different question order compared to the English version.

**(12) MCCB (MATRICS Consensus Cognitive Battery).** A comprehensive, clinician-administered battery of tests designed to measure cognitive functioning in individuals with schizophrenia and other psychiatric disorders. The scale content is available at MCCB.

**(13) YMRS (Young Mania Rating Scale).** A clinician-rated scale used to assess the severity of manic symptoms in individuals with bipolar disorder. The scale content is available at YMRS.

### B.2.2 Relevant Question Selection

We calculated the correlation scores between each question and the total score across the 13 scales mentioned above, as well as their correlation with the presence of mood disorders, using a training set of 2,243 cases. The results are presented below:

| Question ID | Correlation Score | PC p-value |
|---|---|---|
| *1. CTQ* | | |
| ctq_Q1 | 0.1771 | 3.9e-17 |
| ctq_Q2 | 0.1259 | 2.5e-09 |
| ctq_Q3 | 0.2428 | 3.1e-31 |
| ctq_Q4 | 0.2000 | 1.6e-21 |
| ctq_Q5 | 0.0914 | 1.6e-05 |
| ctq_Q6 | 0.1797 | 1.3e-17 |
| ctq_Q7 | 0.2371 | 8.0e-30 |
| ctq_Q8 | 0.4358 | 7.5e-104 |
| ctq_Q9 | 0.1150 | 5.3e-08 |
| ctq_Q10 | 0.3248 | 7.6e-56 |
| ctq_Q11 | 0.2805 | 1.6e-41 |
| ctq_Q12 | 0.2726 | 3.1e-39 |
| ctq_Q13 | 0.2751 | 6.2e-40 |
| ctq_Q14 | 0.3973 | 4.6e-85 |
| ctq_Q15 | 0.2841 | 1.4e-42 |
| ctq_Q16 | -0.2860 | 3.7e-43 |
| ctq_Q17 | 0.1306 | 6.3e-10 |
| ctq_Q18 | 0.3640 | 1.0e-70 |
| ctq_Q19 | 0.2959 | 3.2e-46 |
| ctq_Q20 | 0.1276 | 1.5e-09 |
| ctq_Q21 | 0.0810 | 1.3e-04 |
| ctq_Q22 | -0.2533 | 6.3e-34 |
| ctq_Q23 | 0.1246 | 3.7e-09 |
| ctq_Q24 | 0.1690 | 9.9e-16 |
| ctq_Q25 | 0.3855 | 8.7e-80 |
| ctq_Q26 | 0.1815 | 6.3e-18 |
| ctq_Q27 | 0.0478 | 2.4e-02 |
| ctq_Q28 | 0.3101 | 8.0e-51 |
| *2. DAS* | | |
| das_Q1 | 0.2455 | 6.9e-32 |
| das_Q2 | 0.0296 | 1.6e-01 |
| das_Q3 | 0.2930 | 2.7e-45 |
| das_Q4 | 0.3332 | 8.1e-59 |
| das_Q5 | 0.2801 | 2.2e-41 |
| das_Q6 | 0.1511 | 7.7e-13 |
| das_Q7 | 0.3111 | 4.1e-51 |
| das_Q8 | 0.3161 | 8.1e-53 |
| das_Q9 | 0.3706 | 2.3e-73 |
| das_Q10 | 0.3823 | 2.4e-78 |
| das_Q11 | 0.3618 | 8.6e-70 |
| das_Q12 | 0.1682 | 1.4e-15 |
| das_Q13 | 0.2901 | 2.1e-44 |
| das_Q14 | 0.3695 | 6.1e-73 |
| das_Q15 | 0.3308 | 6.0e-58 |
| das_Q16 | 0.3399 | 2.8e-61 |
| das_Q17 | -0.2876 | 1.2e-43 |
| das_Q18 | -0.0604 | 4.4e-03 |
| das_Q19 | 0.3015 | 5.4e-48 |
| das_Q20 | 0.2773 | 1.4e-40 |
| das_Q21 | 0.1974 | 5.6e-21 |
| das_Q22 | 0.2228 | 2.0e-26 |
| das_Q23 | 0.3144 | 3.0e-52 |
| das_Q24 | 0.0429 | 4.3e-02 |
| das_Q25 | -0.0428 | 4.3e-02 |
| das_Q26 | 0.3093 | 1.6e-50 |
| das_Q27 | 0.3224 | 5.7e-55 |
| das_Q28 | 0.2095 | 1.7e-23 |
| das_Q29 | 0.1205 | 1.2e-08 |
| das_Q30 | 0.0003 | 9.9e-01 |
| das_Q31 | 0.3770 | 4.3e-76 |
| das_Q32 | 0.3108 | 4.9e-51 |
| das_Q33 | 0.2986 | 4.8e-47 |
| das_Q34 | 0.2978 | 8.5e-47 |
| das_Q35 | 0.1615 | 1.8e-14 |
| das_Q36 | 0.2307 | 2.9e-28 |
| das_Q37 | 0.2021 | 6.1e-22 |
| das_Q38 | 0.2937 | 1.6e-45 |
| das_Q39 | 0.2404 | 1.2e-30 |
| das_Q40 | 0.2141 | 1.7e-24 |
| das_total_score | 0.4389 | 2.0e-105 |
| *3. GAD-7* | | |
| gad7_Q1 | 0.4377 | 1.1e-104 |
| gad7_Q2 | 0.4492 | 8.3e-111 |
| gad7_Q3 | 0.4393 | 1.6e-105 |
| gad7_Q4 | 0.4522 | 2.0e-112 |
| gad7_Q5 | 0.3891 | 3.3e-81 |
| gad7_Q6 | 0.4661 | 2.9e-120 |
| gad7_Q7 | 0.3493 | 8.9e-65 |
| gad7_total_score | **0.5116** | 1.7e-148 |
| *4. HCL-32* | | |
| hcl32_Q1 | 0.0179 | 4.0e-01 |
| hcl32_Q2 | -0.1972 | 5.9e-21 |
| hcl32_Q3 | -0.2632 | 1.3e-36 |
| hcl32_Q4 | -0.2292 | 6.5e-28 |
| hcl32_Q5 | -0.1799 | 1.2e-17 |
| hcl32_Q6 | -0.1501 | 1.1e-12 |
| hcl32_Q7 | 0.0251 | 2.4e-01 |
| hcl32_Q8 | 0.2175 | 3.1e-25 |
| hcl32_Q9 | -0.0151 | 4.8e-01 |
| hcl32_Q10 | -0.2148 | 1.2e-24 |
| hcl32_Q11 | -0.1509 | 8.2e-13 |
| hcl32_Q12 | -0.2202 | 7.4e-26 |
| hcl32_Q13 | -0.2187 | 1.7e-25 |
| hcl32_Q14 | -0.0594 | 5.0e-03 |
| hcl32_Q15 | -0.2313 | 2.0e-28 |
| hcl32_Q16 | 0.0177 | 4.0e-01 |
| hcl32_Q17 | -0.0184 | 3.9e-01 |
| hcl32_Q18 | -0.1769 | 4.2e-17 |
| hcl32_Q19 | -0.2283 | 1.0e-27 |
| hcl32_Q20 | -0.1844 | 1.8e-18 |
| hcl32_Q21 | 0.1239 | 4.5e-09 |
| hcl32_Q22 | -0.2486 | 1.0e-32 |
| hcl32_Q23 | 0.0968 | 4.7e-06 |
| hcl32_Q24 | -0.1985 | 3.2e-21 |
| hcl32_Q25 | 0.2063 | 8.0e-23 |
| hcl32_Q26 | 0.1971 | 6.2e-21 |
| hcl32_Q27 | 0.1479 | 2.3e-12 |
| hcl32_Q28 | -0.2751 | 6.2e-40 |
| hcl32_Q29 | 0.0465 | 2.8e-02 |
| hcl32_Q30 | -0.0205 | 3.3e-01 |
| hcl32_Q31 | 0.0117 | 5.8e-01 |
| hcl32_Q32 | 0.0905 | 1.9e-05 |
| hcl32_total_score | -0.0788 | 2.0e-04 |
| *5. MDQ* | | |
| mdq_Q1 | 0.1832 | 3.0e-18 |
| mdq_Q2 | 0.2916 | 7.3e-45 |
| mdq_Q3 | -0.1594 | 3.9e-14 |
| mdq_Q4 | 0.2980 | 7.4e-47 |
| mdq_Q5 | 0.0515 | 1.5e-02 |

| Question ID | Correlation Score | PC p-value |
|---|---|---|
| mdq_Q6 | -0.0556 | 8.8e-03 |
| mdq_Q7 | 0.2141 | 1.8e-24 |
| mdq_Q8 | -0.1568 | 1.0e-13 |
| mdq_Q9 | -0.1437 | 9.7e-12 |
| mdq_Q10 | 0.0857 | 5.2e-05 |
| mdq_Q11 | 0.0907 | 1.8e-05 |
| mdq_Q12 | 0.2022 | 5.8e-22 |
| mdq_Q13 | 0.2920 | 5.6e-45 |
| mdq_total_score | 0.1742 | 1.3e-16 |
| *6. NSSI* | | |
| nssi_Q1 | 0.4597 | 4.0e-116 |
| nssi_Q2 | 0.3568 | 1.9e-67 |
| nssi_Q3 | 0.0221 | 3.0e-01 |
| nssi_Q4 | 0.1636 | 9.5e-15 |
| nssi_Q5 | 0.4422 | 1.4e-106 |
| nssi_Q6 | 0.2471 | 3.8e-32 |
| nssi_Q7 | 0.3085 | 5.4e-50 |
| nssi_Q8 | 0.3112 | 6.5e-51 |
| nssi_Q9 | 0.4045 | 6.8e-88 |
| nssi_Q10 | 0.3417 | 1.3e-61 |
| nssi_Q11 | 0.2181 | 3.1e-25 |
| nssi_Q12 | 0.2392 | 3.7e-30 |
| nssi_Q13 | 0.2664 | 2.8e-37 |
| nssi_Q14 | 0.2641 | 1.3e-36 |
| nssi_Q15 | 0.0729 | 6.0e-04 |
| nssi_Q16 | -0.0825 | 1.0e-04 |
| nssi_Q17 | 0.1995 | 2.7e-21 |
| nssi_Q18 | 0.0267 | 2.1e-01 |
| *7. PHQ-9* | | |
| phq9_Q1 | 0.5008 | 9.0e-143 |
| phq9_Q2 | 0.5006 | 1.2e-142 |
| phq9_Q3 | 0.4842 | 3.2e-132 |
| phq9_Q4 | **0.5097** | 1.2e-148 |
| phq9_Q5 | 0.4484 | 1.9e-111 |
| phq9_Q6 | 0.4877 | 1.9e-134 |
| phq9_Q7 | 0.4578 | 1.1e-116 |
| phq9_Q8 | 0.4493 | 6.6e-112 |
| phq9_Q9 | **0.5057** | 5.4e-146 |
| phq9_total_score | **0.5920** | 2.7e-212 |
| *8. SHAPS* | | |
| shaps_Q1 | 0.1860 | 8.4e-19 |
| shaps_Q2 | 0.2637 | 8.5e-37 |
| shaps_Q3 | 0.2561 | 1.0e-34 |
| shaps_Q4 | 0.2186 | 1.6e-25 |
| shaps_Q5 | 0.1925 | 4.7e-20 |
| shaps_Q6 | 0.1774 | 3.1e-17 |
| shaps_Q7 | 0.2355 | 1.8e-29 |
| shaps_Q8 | 0.1430 | 1.2e-11 |
| shaps_Q9 | 0.2753 | 4.7e-40 |
| shaps_Q10 | 0.1845 | 1.6e-18 |
| shaps_Q11 | 0.2522 | 1.1e-33 |
| shaps_Q12 | 0.2314 | 1.7e-28 |
| shaps_Q13 | 0.1941 | 2.3e-20 |
| shaps_Q14 | 0.1587 | 4.7e-14 |
| shaps_total_score | 0.2675 | 7.4e-38 |
| *9. BPRS* | | |
| bprs_Q1 | 0.1675 | 1.4e-15 |
| bprs_Q2 | 0.4255 | 2.2e-99 |
| bprs_Q3 | 0.1035 | 8.9e-07 |
| bprs_Q4 | 0.0232 | 2.7e-01 |

| Question ID | Correlation Score | PC p-value |
|---|---|---|
| bprs_Q5 | 0.4761 | 2.7e-127 |
| bprs_Q6 | 0.1560 | 1.1e-13 |
| bprs_Q7 | 0.0003 | 9.9e-01 |
| bprs_Q8 | 0.1869 | 4.3e-19 |
| bprs_Q9 | **0.6200** | 1.8e-238 |
| bprs_Q10 | 0.1341 | 1.8e-10 |
| bprs_Q11 | 0.3040 | 3.4e-49 |
| bprs_Q12 | 0.1088 | 2.4e-07 |
| bprs_Q13 | 0.3225 | 1.8e-55 |
| bprs_Q14 | 0.0363 | 8.5e-02 |
| bprs_Q15 | 0.0224 | 2.9e-01 |
| bprs_Q16 | 0.0859 | 4.6e-05 |
| bprs_Q17 | 0.2906 | 6.5e-45 |
| bprs_Q18 | -0.0506 | 1.7e-02 |
| bprs_total_score | 0.4724 | 4.2e-125 |
| *10. HAMA* | | |
| hama_Q1 | 0.4227 | 6.2e-98 |
| hama_Q2 | 0.4177 | 1.9e-95 |
| hama_Q3 | 0.2960 | 1.3e-46 |
| hama_Q4 | **0.5099** | 8.7e-149 |
| hama_Q5 | 0.4987 | 2.2e-141 |
| hama_Q6 | **0.6304** | 7.4e-249 |
| hama_Q7 | 0.2840 | 6.9e-43 |
| hama_Q8 | 0.2837 | 8.2e-43 |
| hama_Q9 | 0.3875 | 2.7e-81 |
| hama_Q10 | 0.3994 | 1.0e-86 |
| hama_Q11 | 0.3641 | 2.7e-71 |
| hama_Q12 | 0.1479 | 1.9e-12 |
| hama_Q13 | 0.2420 | 2.8e-31 |
| hama_Q14 | 0.1529 | 3.3e-13 |
| hama_total_score | **0.5513** | 1.2e-178 |
| *11. HAMD* | | |
| hamd_Q1 | **0.6021** | 8.1e-219 |
| hamd_Q2 | 0.4958 | 6.5e-138 |
| hamd_Q3 | **0.6024** | 4.1e-219 |
| hamd_Q4 | 0.5247 | 4.3e-157 |
| hamd_Q5 | 0.4385 | 7.2e-105 |
| hamd_Q6 | 0.3932 | 6.9e-83 |
| hamd_Q7 | **0.6070** | 2.8e-223 |
| hamd_Q8 | 0.3943 | 2.4e-83 |
| hamd_Q9 | 0.3444 | 9.0e-63 |
| hamd_Q10 | 0.4053 | 2.1e-88 |
| hamd_Q11 | 0.4846 | 6.5e-131 |
| hamd_Q12 | 0.4143 | 1.1e-92 |
| hamd_Q13 | 0.4448 | 3.3e-108 |
| hamd_Q14 | 0.1405 | 3.0e-11 |
| hamd_Q15 | 0.1536 | 3.6e-13 |
| hamd_Q16 | 0.1968 | 8.5e-21 |
| hamd_Q17 | 0.2599 | 1.5e-35 |
| hamd_Q18 | 0.2671 | 1.5e-37 |
| hamd_Q19 | 0.3108 | 7.6e-51 |
| hamd_Q20 | 0.2943 | 1.6e-45 |
| hamd_Q21 | 0.2604 | 1.1e-35 |
| hamd_Q22 | **0.5236** | 2.3e-156 |
| hamd_Q23 | 0.4595 | 3.2e-116 |
| hamd_Q24 | 0.4293 | 3.9e-100 |
| hamd_total_score | **0.6348** | 2.2e-250 |
| *12. MCCB* | | |
| TMT-A | 0.2300 | 4.6e-28 |
| TMT-B | 0.1530 | 4.2e-13 |

| Question ID | Correlation Score | PC p-value |
|---|---|---|
| BACS SC | -0.2589 | 2.3e-35 |
| HVLT-R.1 | -0.0672 | 1.5e-03 |
| HVLT-R.2 | 0.0157 | 4.6e-01 |
| HVLT-R.3 | 0.0536 | 1.1e-02 |
| WMS-III SS | -0.1140 | 7.1e-08 |
| NAB Mazes | -0.1402 | 3.1e-11 |
| BVMT-R.1 | -0.0217 | 3.1e-01 |
| BVMT-R.2 | -0.0255 | 2.3e-01 |
| BVMT-R.3 | -0.0510 | 1.6e-02 |
| Fluency | -0.0489 | 2.1e-02 |
| MSCEIT ME | -0.2084 | 3.2e-23 |
| CPT-IP.1 | -0.2915 | 8.8e-45 |
| CPT-IP.2 | -0.3012 | 8.2e-48 |
| CPT-IP.3 | -0.2747 | 9.2e-40 |
| *13. YMRS* | | |
| ymrs_Q1 | 0.2262 | 1.1e-20 |
| ymrs_Q2 | 0.2100 | 5.3e-18 |
| ymrs_Q3 | 0.0520 | 3.4e-02 |
| ymrs_Q4 | 0.1925 | 2.6e-15 |
| ymrs_Q5 | 0.3067 | 1.9e-37 |
| ymrs_Q6 | 0.2548 | 5.3e-26 |
| ymrs_Q7 | 0.1239 | 4.1e-07 |
| ymrs_Q8 | 0.1106 | 6.4e-06 |
| ymrs_Q9 | 0.1501 | 8.0e-10 |
| ymrs_Q10 | -0.0730 | 2.9e-03 |
| ymrs_Q11 | 0.0900 | 2.4e-04 |
| ymrs_total_score | 0.3196 | 1.0e-40 |

To ensure the selection of highly relevant questions, we identified the top 5% of questions with the highest correlation coefficients. The distribution of correlation scores and the selection threshold are illustrated in Figure 2. Question IDs with correlation scores above the threshold are highlighted in purple, and their corresponding correlation scores are bolded for emphasis. Upon reviewing the original questions selected, we observed that they naturally clustered into five distinct categories: five questions related to depressive mood, two related to suicidal tendencies, three related to loss of interest, two related to anxiety, and two related to insomnia.

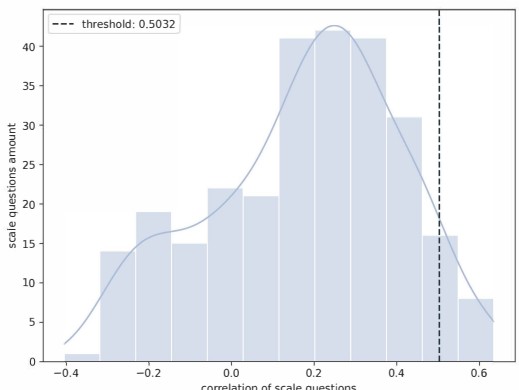

Figure 2: The correlation distribution and threshold of related questions.

Notably, for questions related to depressive mood and loss of interest, we observed variations in how similar questions were phrased across different scales. To ensure robustness, we consulted our domain expert co-author and decided to include two additional questions from the self-reported PHQ-9 depression scale: phq9_Q2 (depressed mood over the past two weeks) and phq9_Q1 (loss of interest over the past two weeks). We highlight these two question IDs in teal. Although their correlation scores (0.5006 and 0.5008, respectively) were slightly below the threshold of 0.5032,

their inclusion provides valuable insights and enhances the diagnostic framework. This decision underscores the importance of multi-scale assessment in minimizing misjudgments of clients' mental states, reducing the impact of short-term emotional fluctuations, and mitigating potential inaccuracies due to inconsistent responses. By repeating similar questions across self-reported and clinician-rated scales, we can achieve a more comprehensive and reliable evaluation. The detailed content and options of selected questions are listed in Table 7.

Table 7: Question content and options of selected scale questions.

| Question ID | Correlation | Question Content | Options |
|---|---|---|---|
| *Depression-related Performances* | | | |
| hamd_total_score | 0.6348 | The total score of Hamilton Depression Rating Scale (HAMD). | 0-6 = no depression, 7-16 = may have depression, 17-23 = must have depression, 24-76 = severe depression |
| hama_Q6 | 0.6304 | Depressed mood. Loss of interest, lack of pleasure in hobbies, depression, early waking, diurnal swing. | 0 = Not present, 1 = Mild, 2 = Moderate, 3 = Severe, 4 = Very severe. |
| bprs_Q9 | 0.6200 | DEPRESSIVE MOOD. Despondency in mood, sadness. Rate only degree of despondency; do not rate on the basis of inferences concerning depression based upon general retardation and somatic complaints. | 0 = not assessed, 1 = not present, 2 = very mild, 3 = mild, 4 = moderate, 5 = moderately severe, 6 = severe, 7 = extremely severe |
| hamd_Q1 | 0.6021 | DEPRESSED MOOD (sadness, hopeless, helpless, worthless) | 0 = Absent. 1 = These feeling states indicated only on questioning. 2 = These feeling states spontaneously reported verbally. 3 = Communicates feeling states non-verbally, i.e. through facial expression, posture, voice and tendency to weep. 4 = Patient reports virtually only these feeling states in his/her spontaneous verbal and non-verbal communication. |
| phq9_total_score | 0.592 | The total score of Patient Health Questionnaire-9 (PHQ-9). | 0-4 = minimal depression, 5-9 = mild depression, 10-14 = moderate depression, 15-19 = moderately severe depression, 20-27 = severe depression |
| phq9_Q2 | 0.5006 | Over the last 2 weeks, how often have you been bothered by any of the following problems? Feeling down, depressed, or hopeless. | 0 = Not at all, 1= Several days, 2 = More than half the days, 3 = Nearly every day |
| *Suicide-related Performances* | | | |
| hamd_Q3 | 0.6024 | SUICIDE | 0 = Absent. 1 = Feels life is not worth living. 2 = Wishes he/she were dead or any thoughts of possible death to self. 3 = Ideas or gestures of suicide. 4 = Attempts at suicide (any serious attempt rate 4). |
| phq9_Q9 | 0.5007 | Over the last 2 weeks, how often have you been bothered by any of the following problems? Thoughts that you would be better off dead, or of hurting yourself. | 0 = Not at all, 1= Several days, 2 = More than half the days, 3 = Nearly every day |
| *Energy&Interest-related Performances* | | | |
| hamd_Q7 | 0.607 | Work and Activities | 0 = No difficulty, 1= Thoughts and feelings of incapacity, fatigue, or weakness related to activities, work, or hobbies, only reported when asked, 2 = Spontaneously reports loss of interest in activities, work, or hobbies, either directly or indirectly, such as feeling listless, indecisive, or needing to push themselves to work or engage in activities, 3 = Decrease in actual time spent in activities or decrease in productivity; in a hospital setting, rate 3 if the patient does not spend at least three hours a day in activities, exclusive of ward chores, 4 = Stopped working due to the current illness; in a hospital setting, rate 4 if the patient engages in no activities except ward chores or if the patient fails to perform ward chores unassisted. |
| hamd_Q22 | 0.5236 | Feelings of Inadequacy or Reduced Ability | 0 = Absent, 1 = Subjective feelings of inadequacy only elicited on questioning, 2 = Patient spontaneously reports feelings of inadequacy), 3 = Needs encouragement, guidance, and reassurance to complete daily tasks or personal hygiene, 4 = Requires assistance from others for dressing, grooming, eating, making the bed, or personal hygiene. |
| phq9_Q4 | 0.5097 | Over the last 2 weeks, how often have you been bothered by any of the following problems? Feeling tired or having little energy. | 0 = Not at all, 1= Several days, 2 = More than half the days, 3 = Nearly every day |
| phq9_Q1 | 0.5008 | Over the last 2 weeks, how often have you been bothered by any of the following problems? Little interest or pleasure in doing things. | 0 = Not at all, 1 = Several days, 2 = More than half the days, 3 = Nearly every day |
| *Anxiety-related Performances* | | | |
| hama_total_score | 0.5513 | The total score of Hamilton Anxiety Rating Scale (HAM-A). | 0-6 = no anxiety, 7-13 = may have anxiety, 14-20 = must have anxiety, 21-28 = must have obvious anxiety, 29-56 = severe anxiety |
| gad7_total | 0.5185 | The total score of Generalized Anxiety Disorder-7(GAD-7). | 0-4 = no anxiety, 5-9 = mild anxiety, 10-14 = moderate anxiety, 15-21 = severe anxiety |
| *Insomnia-related Performances* | | | |
| hamd_Q4 | 0.5247 | INSOMNIA: EARLY IN THE NIGHT | 0 = No difficulty falling asleep. 1 = Complains of occasional difficulty falling asleep, i.e. more than half an hour. 2 = Complaints of nightly difficulty falling asleep. |
| hama_Q4 | 0.5099 | Insomnia. Difficulty in falling asleep, broken sleep, unsatisfying sleep and fatigue on waking, dreams, nightmares, night terrors. | 0 = Not present, 1 = Mild, 2 = Moderate, 3 = Severe, 4 = Very severe. |

## B.3 Retrieval Datastore

### B.3.1 Mental Disorders

In this section, we provide a brief overview of 18 common mental disorders, along with examples of their typical symptoms as outlined in the DSM-5. It is important to note that mental disorders are complex, and their manifestations often extend beyond the descriptions provided here. The following content is intended to offer a general understanding of these conditions and should not be considered exhaustive or definitive.

**Neurodevelopmental Disorders.** The neurodevelopmental disorders are a group of conditions with onset in the developmental period. The range of developmental deficits varies from very specific limitations of learning or control of executive functions to global impairments of social skills or intelligence. For example, individuals with neurodevelopmental disorders may have difficulties in speech and language development, problems with social communication and understanding social cues, repetitive behaviors, and restricted interests.

**Schizophrenia Spectrum and Other Psychotic Disorders.** Schizophrenia spectrum disorders are characterized by a range of symptoms that affect thinking, perception, emotional regulation, and behavior. For example, individuals with auditory hallucinations would hear voices that others do not.

**Bipolar and Related Disorders.** Bipolar and related disorders are characterized by significant mood swings that include episodes of mania or hypomania (elevated mood) and depression. For example, individuals with mania may experience an abnormally elevated, expansive, or irritable mood, increased energy, reduced need for sleep, grandiosity, impulsivity, and excessive engagement in risky behaviors.

**Depressive Disorders.** Depressive disorders are characterized by persistent feelings of sadness, hopelessness, and a loss of interest or pleasure in most activities. For example, individuals with depression may experience changes in appetite or weight, sleep disturbances (either insomnia or excessive sleeping), fatigue, feelings of worthlessness or excessive guilt, difficulty concentrating, and thoughts of death or suicide.

**Anxiety Disorders.** Anxiety disorders are characterized by excessive fear, worry, or nervousness that is disproportionate to the actual threat or situation. For example, individuals with generalized anxiety disorder (GAD) experience chronic, uncontrollable worry about various aspects of life, such as work, health, or social interactions.

**Obsessive-Compulsive and Related Disorders** Obsessive-Compulsive and Related Disorders are characterized by the presence of obsessions (intrusive, unwanted thoughts) and/or compulsions (repetitive behaviors or mental acts performed to reduce anxiety). For example, individuals with Obsessive-Compulsive Disorder (OCD) experience persistent, distressing obsessions and feel compelled to perform rituals or routines to alleviate the anxiety caused by these thoughts.

**Trauma- and Stressor-Related Disorders.** Trauma- and Stressor-Related Disorders are characterized by the emotional and psychological response to traumatic or stressful events. For example, individuals with Post-Traumatic Stress Disorder (PTSD) intrusive memories of the trauma (flashbacks, nightmares), emotional numbness, avoidance of reminders of the event, hypervigilance, and heightened arousal, such as irritability, difficulty sleeping, and exaggerated startle responses.

**Dissociative Disorders.** Dissociative disorders are characterized by disruptions or breakdowns in memory, consciousness, identity, or perception. For example, individuals with Dissociative Identity Disorder (DID), previously known as Multiple Personality Disorder, exhibit two or more distinct identities or personality states, each with its own pattern of thinking, feeling, and behaving. These identities may take control of the person's behavior and are often accompanied by gaps in memory or awareness of time.

**Somatic Symptom and Related Disorders** Somatic Symptom and Related Disorders are characterized by the presence of physical symptoms that cause significant distress or impairment, which are not fully explained by a medical condition. For example, individuals with Illness Anxiety Disorder (formerly known as hypochondriasis) may experience excessive worry about having or developing a serious illness, despite having little or no physical symptoms.

**Feeding and Eating Disorders** Feeding and Eating Disorders are characterized by persistent disturbances in eating behaviors and related thoughts or emotions that negatively impact physical health, emotional well-being, and daily functioning. For example, individuals with Anorexia Nervosa have an intense fear of gaining weight and engage in restrictive eating, leading to significantly low body weight. They may also have a distorted body image, perceiving themselves as overweight even when underweight.

**Elimination Disorders** Elimination Disorders are characterized by the inappropriate elimination of urine or feces, which are not due to a medical condition. For example, individuals with enuresis refers to the repeated involuntary or intentional voiding of urine, typically at night (bedwetting), beyond the expected age for bladder control.

**Sleep-Wake Disorders** Sleep-Wake Disorders are characterized by disturbances in the quality, timing, and amount of sleep, leading to significant impairment in daytime functioning and distress. These disorders are not attributable to the physiological effects of a substance or another medical condition. For example, individuals with insomnia disorder experience persistent difficulty falling asleep, staying asleep, or achieving restorative sleep, despite adequate opportunity for sleep, resulting in fatigue, mood disturbances, and impaired cognitive or social functioning.

**Sexual Dysfunctions** Sexual Dysfunctions are characterized by a clinically significant inability to participate in or experience satisfaction from sexual activity, often causing marked distress or interpersonal difficulties. These dysfunctions are not better explained by a nonsexual mental disorder, relationship distress, or the effects of a substance or medical condition. For example, individuals with erectile disorder experience a persistent or recurrent inability to attain or maintain an adequate erection during sexual activity, leading to significant distress or interpersonal strain.

**Gender Dysphoria** Gender Dysphoria involves a strong and persistent discomfort with one's assigned gender at birth and a desire to be treated as the opposite gender. This condition is marked by significant distress or impairment in functioning due to the incongruence between experienced or expressed gender and assigned gender. For example, a person assigned female at birth may experience intense discomfort with their body, wishing to transition to a male gender identity, often leading to emotional and social distress.

**Disruptive, Impulse-Control, and Conduct Disorders** These disorders are characterized by persistent patterns of behavior where the rights of others or societal norms are violated. Individuals with these disorders often exhibit aggressive, antisocial, or impulsive behaviors that disrupt their social, academic, or occupational functioning. For example, a child with conduct disorder may frequently engage in theft, aggression toward others, or deliberate property destruction, showing little empathy or remorse for their actions.

**Substance-Related and Addictive Disorders** Substance-Related and Addictive Disorders encompass a range of problems caused by the use of substances (e.g., alcohol, drugs) or behaviors (e.g., gambling) that lead to addiction. These disorders are defined by a pattern of substance use or behavior that leads to significant impairment, including physical, social, or psychological problems. For example, an individual with alcohol use disorder may find themselves drinking excessively despite negative consequences, such as relationship issues or health problems, and may experience withdrawal symptoms when not drinking.

**Neurocognitive Disorders** Neurocognitive Disorders involve a decline in cognitive function that represents a significant change from a previous level of functioning. These disorders can affect memory, learning, attention, executive function, and perception. For example, Alzheimer's disease is a common neurocognitive disorder where individuals experience progressive memory loss and confusion, often leading to difficulty performing everyday tasks.

**Personality Disorders** Personality Disorders are characterized by enduring patterns of behavior, cognition, and inner experience that deviate markedly from the expectations of an individual's culture. These patterns are inflexible and pervasive, leading to distress or impairment in social, occupational, or other important areas of functioning. For example, individuals with borderline personality disorder may have intense and unstable relationships, fear of abandonment, and difficulty regulating their emotions, often leading to impulsive actions or self-harming behaviors.

### B.3.2   Knowledge Base Construction

To gather professional diagnostic criteria, we rely on the *Diagnostic and Statistical Manual of Mental Disorders: DSM-5* [15], a comprehensive and authoritative guide widely used by clinicians and researchers. The DSM-5 outlines the standardized criteria for the classification and diagnosis of mental disorders, providing detailed descriptions of symptoms, diagnostic features, and associated conditions. As a crucial tool in psychiatry, it ensures consistency and accuracy in mental health diagnosis, making it an essential reference for both clinical practice and research.

Our knowledge base is built by extracting diagnostic criteria and symptoms from the DSM-5, focusing on mood disorders and other conditions. We reference the *diagnostic criteria* section to identify required symptoms and their severity, and the *differential diagnosis* section to distinguish mood disorders from similar conditions. This ensures accurate diagnosis while addressing symptom overlaps.

**Diagnostic Criteria Extraction.** Symptoms in the DSM-5 are listed individually. To facilitate comparison with client records, we reframe these using GPT-4o for brevity and clarity, ensuring independent, complete criteria. We extracted symptoms for 18 common mental disorders, including all major symptoms of "bipolar and related disorders" and "depressive disorders." The extraction prompt is provided in Table 8.

Table 8: Prompts of diagnostic criteria extraction.

> To facilitate comparison between diagnostic criteria and medical records, please summarize the criteria for `{disease name}` from the DSM-5 diagnostic manual in a point-by-point format. Each point should fully describe a symptom, explicitly include reference information where necessary, and retain age or other relevant restrictions.
> Below is the description of the diagnostic criteria for `{disease name}` from the DSM-5: `{the original criteria}`.

For example, one symptom of bipolar disorder is as follows:

"A distinct period of abnormally and persistently elevated, expansive, or irritable mood and abnormally and persistently increased goal-directed activity or energy, lasting at least 1 week and present most of the day, nearly every day (or any duration if hospitalization is necessary)."

The extracted symptom description would be:

"Manic episode: Elevated, expansive, or irritable mood, accompanied by increased energy and activity, lasting at least 1 week and present most of the day, nearly every day."

**Differential Criteria Extraction.** We decompose complex differential diagnosis into distinct expressions using GPT-4o. This process ensures precise differentiation between mood disorder and other diseases. The extraction prompt is provided in Table 9.

Table 9: Prompts of differential criteria extraction.

> Please split the following differential criteria into two separate diagnostic criteria. Each point should fully describe a symptom, explicitly include reference information where necessary, and retain age or other relevant restrictions.
> Below are the differential criteria to be split: `the original criteria`.

For example, a differential symptom for bipolar disorder is described as:

"Attention-deficit/hyperactivity disorder. This disorder may be misdiagnosed as bipolar disorder, especially in adolescents and children. Many symptoms overlap with the symptoms of mania, such as rapid speech, racing thoughts, distractibility, and less need for sleep. The 'double counting' of symptoms toward both ADHD and bipolar disorder can be avoided if the clinician clarifies whether the symptom(s) represent a distinct episode."

This description was refined into two distinct expressions:

"Attention-deficit/hyperactivity disorder: rapid speech, racing thoughts, distractibility, and less need for sleep in adolescents and children."

"Bipolar disorder: rapid speech, racing thoughts, distractibility, and less need for sleep in adults."

### B.3.3  Knowledge Base Evaluation

We conducted a comprehensive evaluation of the knowledge base to ensure the accuracy and completeness of the symptom descriptions for all entries, with a particular focus on the sections covering "bipolar and related disorders" and "depressive disorders." The evaluation involved assessing the correctness of each symptom description and verifying the completeness of the content within these two critical sections. Any inaccuracies or gaps identified during this process were manually revised to ensure the highest quality of information.

## C   Agent Details

### C.1   Three Diagnostic Variants

In this section, we provide prompts of Angel.R, Angel.D and Angel.C in Table 10.

Table 10: Prompts of Angel.R, Angel.D and Angel.C.

---

**Role:** You are a psychiatry diagnosis expert.

**Objective:** Diagnose whether the visitor `{digit_id}` has a mood disorder (including depression and bipolar disorder), which is characterized by depressive or manic symptoms.

**Constraints:**

1. You may only use the following actions.

2. You must act proactively. Always keep this in mind when planning actions.

**Available Actions:**

*Three variants of angels are different in actions. Angel.R: 1,2,7; Angel.D: 1,2,3,4,7; Angel.C: 1,2,5,6,7.*

1. toggle_visitor_record: Retrieves structured medical records for the visitor, if available. Also extracts top-5 symptoms related to DSM-5 diagnostic criteria. Args: {"name":"digit_id", "description":"Visitor ID, must be provided in the query", "type":"int"}

2. get_scale_performances: Retrieves the visitor's performance on the top 5% most mood disorder-related questions from psychological scales. The correlation score represents the statistical relevance to mood disorders. Args: {"name":"digit_id", "description":"Visitor ID, must be provided in the query", "type":"int"}

3. previous_cases_display: If the visitor has medical record information, this tool compares the case with past structured medical records in the database and extracts the top 5 most similar cases. Due to the personalized nature of psychiatric diagnosis, the extracted similar cases are for reference only. Args: {"name": "digit_id", "description": "Visitor ID, must be provided in the query", "type": "int"}

4. previous_scales_display: This tool compares the visitor's scale performance with past case scale performances in the database, and extracts the top 5 most similar cases. Due to the personalized nature of psychiatric diagnosis, the extracted similar cases are for reference only. Args: {"name": "digit_id", "description": "Visitor ID, must be provided in the query", "type": "int"}

5. previous_cases_analysis: If the visitor has medical record information, this tool compares the case with past structured medical records in the database, extracts and analyzes the top 5 most similar cases. Due to the personalized nature of psychiatric diagnosis, the extracted similar cases are for reference only. Args: {"name": "digit_id", "description": "Visitor ID, must be provided in the query", "type": "int"}

6. previous_scales_analysis: This tool returns a comparison of the visitor's scale performance with past case scale performances in the database, extracts and analyzes the top 5 most similar cases. Due to the personalized nature of psychiatric diagnosis, the extracted similar cases are for reference only. Args: {"name": "digit_id", "description": "Visitor ID, must be provided in the query", "type": "int"}

7. Finish: Completes the diagnosis process. Args:{"answer":'yes' or 'no' (indicating whether the visitor has a mood disorder)", "reasons":"A summary of key reasons supporting the decision."}

**Resources:** You are a large language model trained on vast textual data, including factual medical knowledge.

**Best Practices:**

1. Always refer to similar cases (medical records or scale data) for diagnosis. However, do not over-rely on past diagnoses due to the personalized nature of mental health conditions.

2. Consider all available information (medical records & scale performance). Always examine similar records (if available) and similar scale data for reference. Do not rely solely on one data source.

3. Prioritize clinical evaluation over self-reported scales in case of contradictions. Self-reported terms like "occasionally", "sometimes", and "frequently" are subjective, making it difficult for visitors to assess their condition accurately.

4. Only consider symptoms relevant to mood disorders. The visitor may have other mental disorders that are not classified as mood disorders.

5. Mood disorders include only depression and bipolar disorder, characterized by depressive or manic symptoms.

**Response Format:**

Generate a JSON string following the structure below. Do not include any extra text or explanation. {"action": {"name": "action name", "args": {"args name": "args value"}}, "thoughts": {"plan": "Briefly describe the list of short-term and long-term plans", "criticism": "Constructive self-criticism", "observation": "Summary of the current step returned to the user", "reasoning": "Reasoning behind the decision"}}

---

### C.2   Symptom Matching

#### C.2.1   Medical Record Processing

To ensure patient privacy, the medical records presented in this section have been anonymized, with all event details adapted while preserving the patient's actual symptoms. The original medical record (adapted) and the corresponding processing steps are illustrated in Figure 3.

Effective psychiatric diagnosis relies on comprehensive and well-structured patient information. However, raw medical records often contain scattered, redundant, or sensitive details that can hinder accurate analysis. To address these challenges, we designed a systematic medical record processing pipeline that ensures data security, enhances temporal reasoning, and improves symptom extraction for LLM-based agents. This pipeline consists of four key steps: extracting essential diagnostic elements, refining present illness descriptions to prevent data leakage, integrating structured information for coherence, and an optimal step of reorganizing records into a structured format for precise symptom matching. The processing prompts are provided in Table 11.

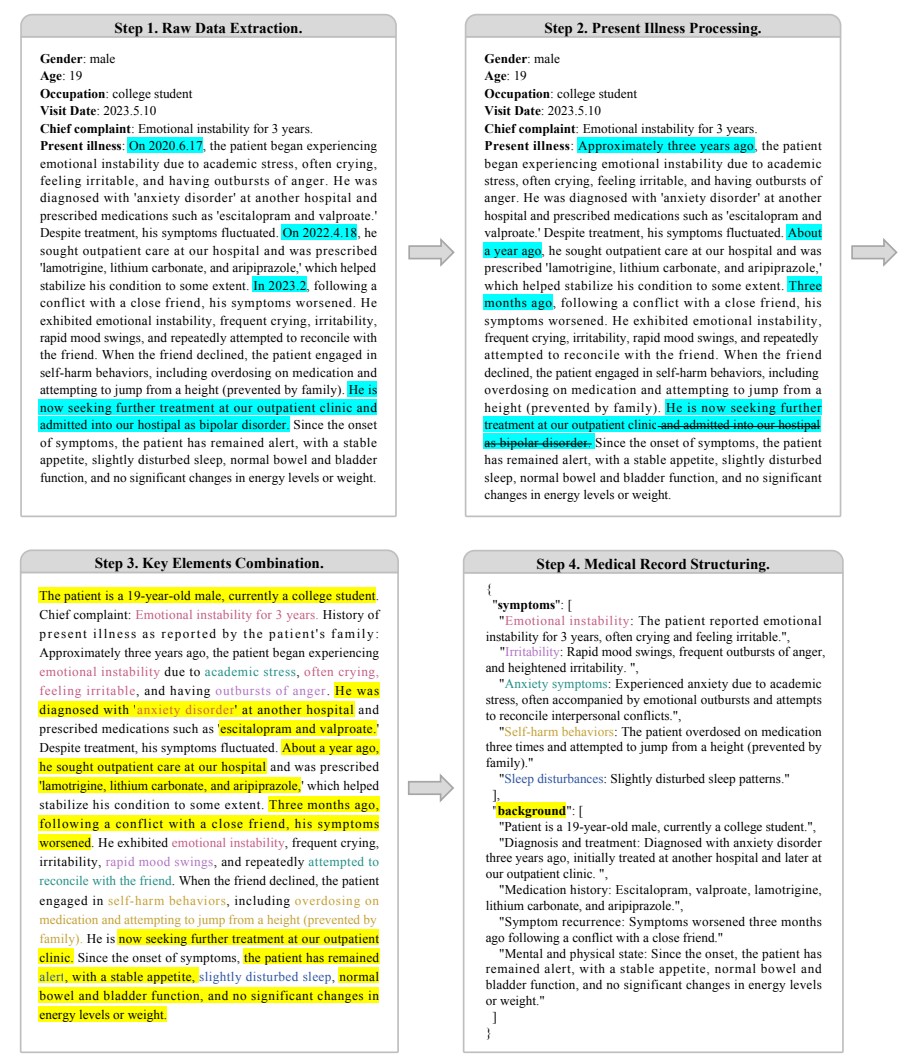

Figure 3: An example of the original medical record (adapted) and its corresponding processing steps.

Table 11: Prompts of medical record processing.

**Step 1 → Step 2:** Replace specific dates in the present illness history with relative time expressions based on the patient's admission date. Avoid using exact dates after the replacement.
**Step 2 → Step 3:** Organize the patient's information into a single, coherent paragraph without altering the content of the present illness section.
**Step 3 → Step 4:** Structure the medical record by extracting and summarizing the patient's symptoms and background. List each item separately, using "-" as a delimiter. Respond only in JSON format.

**Step 1. Raw Data Extraction.** To construct a comprehensive and secure data store, we first obtained anonymized client information from the hospital database. To capture the full context of clients' backgrounds, which are critical for accurate diagnosis, we extracted the following key elements from the medical records: (1) Gender: Essential for identifying gender-specific symptoms (e.g., menstrual-related mood changes). (2) Age: Some disorders manifest at specific life stages, such as adolescence. (3) Occupation: Helps differentiate disorder-related symptoms from external factors (e.g., shift work causing sleep issues). (4) Visit Date: Provides a reference for interpreting time-sensitive information. (5) Chief Complaint: A concise summary of primary symptoms. (6) Present illness: A detailed symptom history, including medication usage.

**Step 2. Present Illness Processing.** To prevent data leakage in agent diagnosis, we removed explicit disorder labels from the present illness section. Additionally, absolute dates were converted into relative timeframes based on the visit date (e.g., "9 months ago" instead of a specific date), enhancing data privacy while facilitating temporal reasoning for symptom progression analysis.

**Step 3. Key Elements Combination.** To generate a coherent input for LLM-based analysis, we integrate the structured elements from Step 1 with the processed present illness data from Step 2, ensuring comprehensive contextual understanding.

**Step 4. Medical Records Structuring.** Symptoms relevant to diagnosis are often dispersed across different parts of medical records. For example, "suspecting others and perceiving malicious intent" and "engaging in defensive behaviors" may appear separately, making direct symptom matching difficult. To address this, we reorganize medical records into a structured format, presenting key symptoms in JSON. We will compare whether coherent text or structured representation is more effective for symptom matching and analysis in the ablation study (Section 4.3.1).

### C.2.2 Symptom Matching Formulation

Symptom matching is to calculate the similarity of embedded medical records and diagnostic criteria. The calculation involves transforming the structured medical record $r$ into hidden states $H_r$ via a text encoder, expressed as $e_r = \text{norm}(H_r[0])$. Similarly, the embedding of each diagnostic criterion $c$ is obtained as $e_c = \text{norm}(H_c[0])$. The similarity score of dense embedding $s_{\text{dense}}$ between the structured record and each criterion is calculated by the inner product of the two embeddings $e_r$ and $e_c$, expressed as $s_{\text{dense}} \leftarrow \langle e_c, e_r \rangle$. This score provides a quantitative measure of the semantic alignment between the client's symptoms and the DSM-5 diagnostic criteria.

### C.3 Scale Performance Analysis

Client performances are converted from numeric scores to textual descriptions based on question content and options, with an example of one question provided in Figure 4, and an example of all related questions is provided in Figure 5.

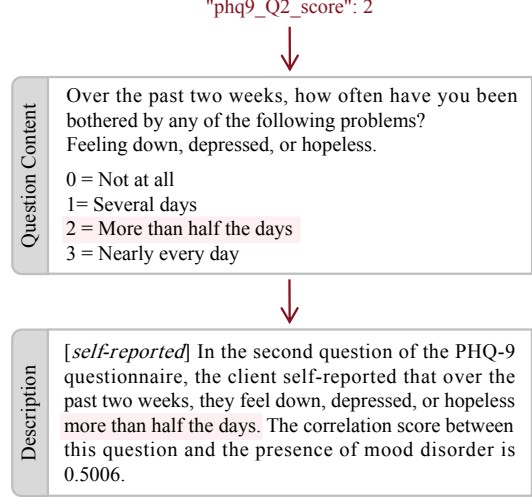

Figure 4: An example of generating scale performance description from the score value of a relevant question.

| | **Scale Performance Descriptions** |
|---|---|
| | The following are the performance of highly relevant questions in the [self-reported] and [clinician-evaluated] scales, as well as the Pearson correlation coefficient between this question and the presence of mood disorders in a statistical sense: |
| | **Depression-related Performances:** |
| "hamd_total_score": 15 | [*clinician-evaluated*] In the Hamilton Depression Rating Scale (HAMD) filled out by the clinician during the consultation, the client scored 15 points (out of 76 points), indicating that the client may have depressive symptoms. The correlation score between this and the presence of mood disorder is 0.6348. |
| "hama_Q6_score": 1 | [*clinician-evaluated*] In the sixth question of the HAMA questionnaire, the clinician assessed the client's depressive mood (Loss of interest, lack of pleasure in hobbies, depression, early waking, diurnal swing.) as mild. The correlation score between this question and the presence of mood disorder is 0.6304. |
| "bprs_Q9_score": 1 | [*clinician-evaluated*] In the ninth question of the BPRS scale, the clinician assessed the client's DEPRESSIVE MOOD (Despondency in mood, sadness. Rate only degree of despondency) as not present. The correlation score between this question and the presence of mood disorder is 0.62. |
| "hamd_Q1_score": 2 | [*clinician-evaluated*] In the first question of the HAMD questionnaire, the clinician assessed the client's depressive mood as: These feeling states spontaneously reported verbally. The correlation score between this question and the presence of mood disorder is 0.6021. |
| "phq9_total_score": 8 | [*self-reported*] In the self-rated Patient Health Questionnaire-9 (PHQ-9), the client scored 8 points (out of a total of 27 points), indicating that the client may have mild depression. The correlation score between this and the presence of mood disorder is 0.592. |
| "phq9_Q2_score": 2 | [*self-reported*] In the second question of the PHQ-9 questionnaire, the client self-reported that over the past two weeks, they feel sad, depressed, or hopeless more than half the days. The correlation score between this question and the presence of mood disorder is 0.5006. |
| | **Suicide-related Performances:** |
| "hamd_Q3_score": 0 | [*clinician-evaluated*] In the third question of the HAMD questionnaire, the clinician assessed the client's suicidal tendencies as: none. The correlation score between this question and the presence of mood disorder is 0.6024. |
| "phq9_Q9_score": 0 | [*self-reported*] In the ninth question of the PHQ-9 questionnaire, the client self-reported that over the past two weeks, they did not at all think that death or harming themselves in some way was a solution. The correlation score between this question and the presence of mood disorder is 0.5057. |
| | **Energy&Interest-related Performances:** |
| "hamd_Q7_score": 1 | [*clinician-evaluated*] In the 7th question of the HAMD questionnaire, the clinician rated the client's work and interests as: Thoughts and feelings of incapacity, fatigue, or weakness related to activities, work, or hobbies, only reported when asked. The correlation score between this question and the presence of mood disorder is 0.607. |
| "hamd_Q22_score": 2 | [*clinician-evaluated*] In the 22nd question of the HAMD questionnaire, the clinician rated the client's Feelings of Inadequacy or Reduced Ability as: patient spontaneously reports feelings of inadequacy. The correlation score between this question and the presence of mood disorder is 0.5236. |
| "phq9_Q4_score": 2 | [*self-reported*] In the 4th question of the PHQ-9 questionnaire, the client self-reported that over the past two weeks, they feel tired or having little energy more than half the days. The correlation score between this question and the presence of mood disorder is 0.5097. |
| "phq9_Q1_score": 2 | [*self-reported*] In the 4th question of the PHQ-9 questionnaire, the client self-reported that over the past two weeks, they have little interest or pleasure in doing things more than half the days. The correlation score between this question and the presence of mood disorder is 0.5008. |
| | **Anxiety-related Performances:** |
| "hama_total_score": 11 | [*clinician-evaluated*] In the Hamilton Anxiety Rating Scale (HAMA) filled out by the doctor during the consultation, the client scored 11 points (out of a total of 56 points), indicating that the client may have anxiety. The correlation score between this question and the presence of mood disorder is 0.5513. |
| "gad7_total_score": 8 | [*self-reported*] In the self-assessed Generalized Anxiety Disorder-7 (GAD-7), the client scored 8 points (out of 21 points), indicating that the client has mild anxiety. The correlation score between this question and the presence of mood disorder is 0.5185. |
| | **Insomnia-related Performances:** |
| "hamd_Q4_score": 0 | [*clinician-evaluated*] In the fourth question of the HAMD questionnaire, the clinician assessed the client's INSOMNIA: EARLY IN THE NIGHT as: No difficulty falling asleep. The correlation score between this question and the presence of mood disorder is 0.5247. |
| "hama_Q4_score": 1 | [*clinician-evaluated*] In the fourth question of the HAMA questionnaire, the clinician assessed the client's Insomnia (Difficulty in falling asleep, broken sleep, unsatisfying sleep and fatigue on waking, dreams, nightmares, night terrors) as mild. The correlation score between this question and the presence of mood disorder is 0.5099. |

Figure 5: A full example of scale performance descriptions derived from the score values of mood disorder-related questions.

## C.4 Multi-agent Diagnosis

In this section, we provide prompts of two debate agents in Table 12 and prompts of the judge in Table 13.

Table 12: Prompts of Debate Agents.

**Role:** You are a psychiatry diagnosis expert. You believe the current visitor `has (does not have)` a mood disorder. Present your argument and persuade the opposing side.
**Visitor Information:** This is a challenging case, and three agents have already provided their diagnoses:
1. The agent without prior case retrieval believes the visitor {`has (does not have)`} a mood disorder. Its reasoning: {`Reasons given by Angel.R`}
2. The agent who retrieved past cases but did not analyze them believes the visitor {`has (does not have)`} a mood disorder. Their reasoning: {`Reasons given by Angel.D`}
3. The agent who retrieved and analyzed past cases believes the visitor {`has (does not have)`} a mood disorder. Their reasoning: {`Reasons given by Angel.C`}
**Original Case Information:**
1. Visitor's Medical Record: {The visitor's medical record in Step 3}
2. Visitor's Scale Performances: {The visitor's performances on selected scales}
3. Reference from Past Medical Records: {Top-5 similar medical records with analysis}
4. Reference from Past Scale Performances: {Top-5 similar scale performances with analysis}
**Constraints:**
1. Stick to your stance. Every case presented for discussion is inherently controversial, meaning your viewpoint is strongly justified.
2. You cannot interact with physical objects. If an action is absolutely necessary, request the user to perform it. If the user refuses and no alternative exists, terminate the process to avoid wasting time and effort.
**Resources:**
1. You are a large language model trained on extensive textual data, including factual medical knowledge.
2. All relevant visitor information is provided. Make full use of it.
**Best Practices:**
1. Consider all available visitor information (medical records & scale performances). Examine both similar medical records and similar scale performances as references. Do not base conclusions on a single source.
2. Due to the personalized nature of psychiatric diagnosis, do not overly rely on similar case diagnoses.
3. Self-reported questionnaire results may contradict clinician evaluations. In such cases, prioritize the clinician's assessment. Terms like "occasionally," "sometimes," and "frequently" in self-reports are subjective and may lead to misinterpretation of the visitor's condition.
4. Only symptoms related to mood disorders should influence your diagnosis. The visitor may have other mental disorders that are not classified as mood disorders.
5. Mood disorders include only depression and bipolar disorder, characterized by depressive or manic symptoms.
**Response Format:**
Generate a JSON string in the following format. Do not include any extra text or explanation.
{"response": "Your reasoning for why the visitor has (or does not have) a mood disorder, along with a rebuttal to the opposing argument.", "thoughts": {"plan": "Briefly outline short-term and long-term plans", "criticism": "Constructive self-criticism"}}

### C.4.1 Debate Details

Our debate stage is intentionally designed to simulate the classical four-stage structure commonly used in formal debates, with a minimum of 2 rounds and a maximum of 4. At the beginning of each round, the prompt guides the Pro and Con agents through a structured process:

- Round 1 – Opening Statements: Both Pro and Con agents present their initial diagnostic positions based on the available information.

- Round 2 – Rebuttals: Each agent critically responds to the opponent's arguments, defending their own stance and challenging the other side's reasoning.

- Round 3 – Free Debate (Optional): If the Judge Agent identifies unresolved or ambiguous points after the first two rounds, it initiates a more open-ended exchange to further clarify conflicting perspectives.

- Round 4 – Closing Statements (Optional): If the discussion remains inconclusive or highly contentious, both agents are given one final opportunity to summarize their most compelling arguments in a concise manner.

After each round, the Judge Agent evaluates whether the debate should continue, provides reasoning, and reflects on the arguments. At the end of the debate, the judge delivers a diagnosis with reasons. The example in Figure 6 demonstrates a case where the debate concluded in three rounds.

This flexible structure allows the system to adapt the level of deliberation to the complexity of the case, balancing efficiency with thoroughness in clinical decision-making. The minimum of two rounds ensures that both sides have the opportunity to present and respond to each other's views—this

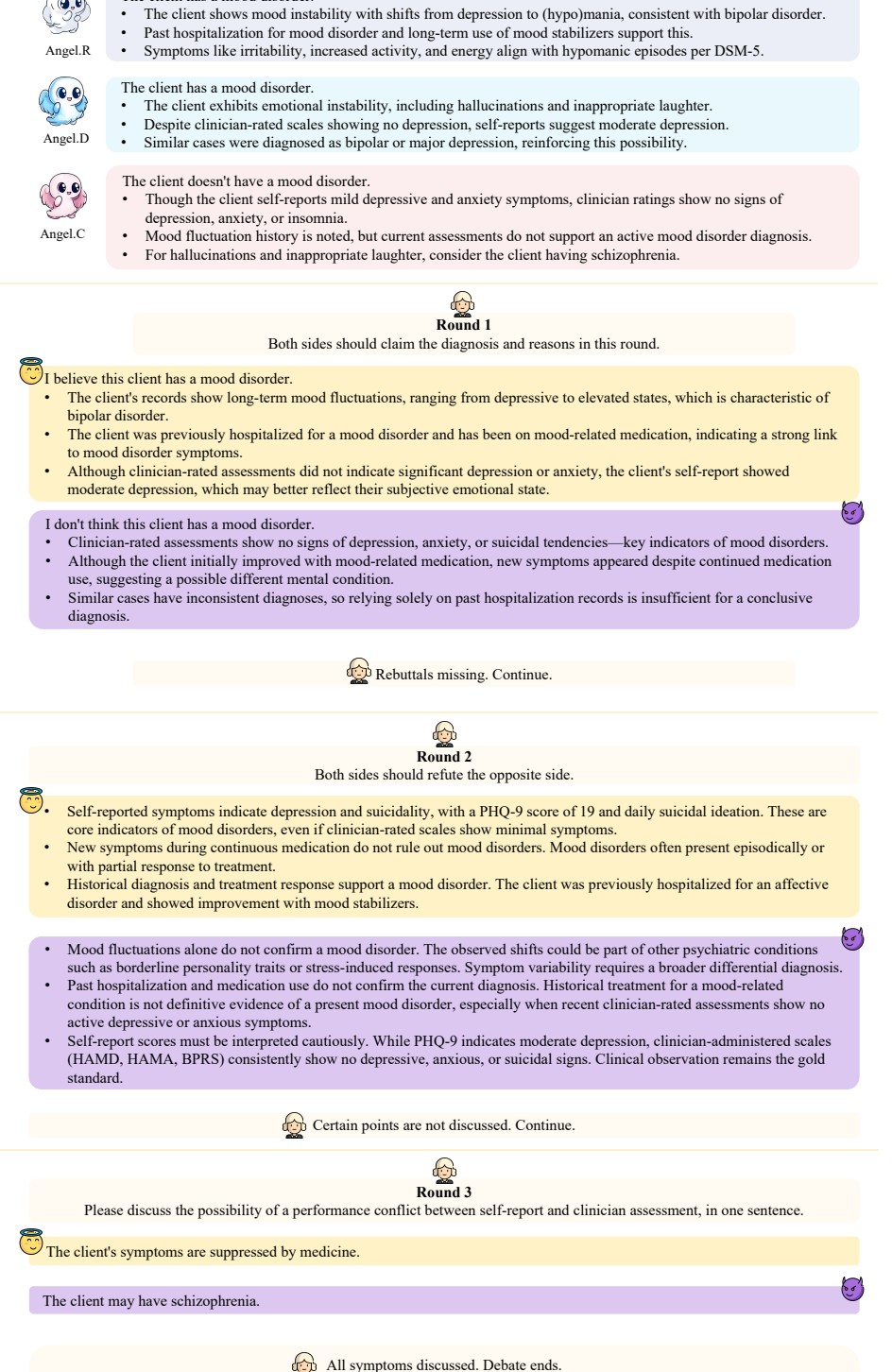

The client has a mood disorder.
- The client shows mood instability with shifts from depression to (hypo)mania, consistent with bipolar disorder.
- Past hospitalization for mood disorder and long-term use of mood stabilizers support this.
- Symptoms like irritability, increased activity, and energy align with hypomanic episodes per DSM-5.

Angel.R

The client has a mood disorder.
- The client exhibits emotional instability, including hallucinations and inappropriate laughter.
- Despite clinician-rated scales showing no depression, self-reports suggest moderate depression.
- Similar cases were diagnosed as bipolar or major depression, reinforcing this possibility.

Angel.D

The client doesn't have a mood disorder.
- Though the client self-reports mild depressive and anxiety symptoms, clinician ratings show no signs of depression, anxiety, or insomnia.
- Mood fluctuation history is noted, but current assessments do not support an active mood disorder diagnosis.
- For hallucinations and inappropriate laughter, consider the client having schizophrenia.

Angel.C

**Round 1**
Both sides should claim the diagnosis and reasons in this round.

I believe this client has a mood disorder.
- The client's records show long-term mood fluctuations, ranging from depressive to elevated states, which is characteristic of bipolar disorder.
- The client was previously hospitalized for a mood disorder and has been on mood-related medication, indicating a strong link to mood disorder symptoms.
- Although clinician-rated assessments did not indicate significant depression or anxiety, the client's self-report showed moderate depression, which may better reflect their subjective emotional state.

I don't think this client has a mood disorder.
- Clinician-rated assessments show no signs of depression, anxiety, or suicidal tendencies—key indicators of mood disorders.
- Although the client initially improved with mood-related medication, new symptoms appeared despite continued medication use, suggesting a possible different mental condition.
- Similar cases have inconsistent diagnoses, so relying solely on past hospitalization records is insufficient for a conclusive diagnosis.

Rebuttals missing. Continue.

**Round 2**
Both sides should refute the opposite side.

- Self-reported symptoms indicate depression and suicidality, with a PHQ-9 score of 19 and daily suicidal ideation. These are core indicators of mood disorders, even if clinician-rated scales show minimal symptoms.
- New symptoms during continuous medication do not rule out mood disorders. Mood disorders often present episodically or with partial response to treatment.
- Historical diagnosis and treatment response support a mood disorder. The client was previously hospitalized for an affective disorder and showed improvement with mood stabilizers.

- Mood fluctuations alone do not confirm a mood disorder. The observed shifts could be part of other psychiatric conditions such as borderline personality traits or stress-induced responses. Symptom variability requires a broader differential diagnosis.
- Past hospitalization and medication use do not confirm the current diagnosis. Historical treatment for a mood-related condition is not definitive evidence of a present mood disorder, especially when recent clinician-rated assessments show no active depressive or anxious symptoms.
- Self-report scores must be interpreted cautiously. While PHQ-9 indicates moderate depression, clinician-administered scales (HAMD, HAMA, BPRS) consistently show no depressive, anxious, or suicidal signs. Clinical observation remains the gold standard.

Certain points are not discussed. Continue.

**Round 3**
Please discuss the possibility of a performance conflict between self-report and clinician assessment, in one sentence.

The client's symptoms are suppressed by medicine.

The client may have schizophrenia.

All symptoms discussed. Debate ends.

The client doesn't have a mood disorder.
Although the client was previously hospitalized and prescribed medication related to mood disorders, multiple clinician-administered scales did not indicate typical symptoms of depression or anxiety. Moreover, the client's current symptoms are not alleviated by mood disorder medications, raising the possibility of other psychiatric conditions with overlapping features, such as schizophrenia.

Figure 6: A debate example where only Angel.C was correct and the other two were wrong, and the Judge made a correct decision by debate.

Table 13: Prompts of the Judge.

Role:
You are a psychiatry diagnosis expert acting as the judge in this consultation. Your role is to determine whether each debate round should conclude and make the final diagnosis.
Visitor Information:
This is a challenging case, and three agents have already provided their diagnoses:
Agent without prior case retrieval believes the visitor {is/isn't, diagnosis by Angel.R} diagnosed with a mood disorder. Their reasoning: {Reasons given by Angel.R}
Agent who retrieved past cases but did not analyze them believes the visitor {is/isn't, diagnosis by Angel.D} diagnosed with a mood disorder. Their reasoning: {Reasons given by Angel.D}
Agent who retrieved and analyzed past cases believes the visitor {is/isn't, diagnosis by Angel.C} diagnosed with a mood disorder. Their reasoning: {Reasons given by Angel.C}
Original Case Information:
Visitor's Medical Record: {The visitor's medical record in Step 3}
Visitor's Scale Performances: {The visitor's performances on selected scales}
Reference from Past Medical Records: {Top-5 similar medical records with analysis}
Reference from Past Scale Performances: {Top-5 similar scale performances with analysis}
Constraints:
Fulfill your role as a judge and base your decision solely on the arguments presented by both sides.
You cannot interact with physical objects. If an action is absolutely necessary to complete the task, request the user to perform it. If the user refuses and no alternative exists, terminate the process to avoid wasting time and effort.
Resources:
You are a large language model trained on extensive textual data, including factual medical knowledge.
All relevant visitor information is provided. Make full use of it.
Best Practices:
Consider all available visitor information (medical records & scale performances). Examine both similar medical records and similar scale performances as references. Do not base conclusions on a single source.
Due to the personalized nature of psychiatric diagnosis, do not overly rely on similar case diagnoses.
Self-reported questionnaire results may contradict clinician evaluations. In such cases, prioritize the clinician's assessment. Terms like "occasionally," "sometimes," and "frequently" in self-reports are subjective and may lead to misinterpretation of the visitor's condition.
Only symptoms related to mood disorders should influence your diagnosis. The visitor may have other mental disorders that are not classified as mood disorders.
Mood disorders include only depression and bipolar disorder, characterized by depressive or manic symptoms.
Response Format:
Generate a JSON string in the following format. Do not include any extra text or explanation.
{"judge": "Do you believe the debate should end? Answer only 'yes' or 'no'.", "diagnose": "Do you believe the visitor has a mood disorder? Answer only 'yes' or 'no'.", "thoughts": { "plan": "Briefly outline short-term and long-term plans.", "criticism": "Constructive self-criticism.", "judge_reasons": "Your reasoning for why the debate should or should not end.", "reasoning": "Your reasoning for whether the visitor does or does not have a mood disorder." }}

establishes a meaningful diagnostic exchange. If the Judge Agent determines that all relevant points have been sufficiently addressed by the end of Round 2, the debate can be concluded early. However, when the case involves ambiguity or conflicting symptom interpretations, the Judge may trigger additional rounds to promote deeper reasoning.

In practice, we found most cases concluded after just two rounds of debate, as the key diagnostic arguments had already been clearly presented and addressed. The remaining cases required only one additional round, with all debates concluding within three rounds. This observation supports the efficiency of our debate framework in ensuring thorough reasoning when needed, while avoiding unnecessary verbosity or repetition.

# D   Synthetic Data

**Usage Declaration.** This work and its contents, including the synthetic mental health diagnostic dataset and diagnostic agent, are provided for academic and research purposes only. None of the material constitutes medical, clinical, or therapeutic advice. The synthetic data is artificially generated and does not represent real patient information. No warranties, express or implied, are offered regarding the accuracy, reliability, or suitability of the dataset, agent, or any related content. The authors and contributors are not responsible for any errors, omissions, or consequences arising from the use of this repository.

This is not a substitute for professional medical or psychological evaluation. Users should consult qualified healthcare providers for clinical assessments or treatment decisions. By using this repository, you acknowledge that the diagnostic agent is experimental and should not guide real-world medical decisions.

The use of the software, data, or any derived materials is entirely at the user's own risk. By accessing or utilizing this repository, you agree to indemnify, defend, and hold harmless the authors, contributors, and affiliated entities from any claims or damages.

## D.1 Synthetic Data Construction

We synthesize our dataset using a structured pipeline that begins with data preparation, followed by model training, and concludes with rigorous post-processing. First, we construct the input data by combining 16 mood disorder-related questions from Table 7, 8 baseline assessment scale total scores, and binary mood disorder labels (1 for presence, 0 for absence), resulting in 25 features per case. This tabular format serves as the foundation for our synthesis process using the TabSyn framework [20].

The training phase consists of two key components. We first pretrain a VAE to encode the tabular data into a continuous latent space, employing column-wise tokenizers and Transformer-based encoders/decoders to handle mixed data types. The model optimizes an adaptively weighted ELBO loss, dynamically balancing reconstruction accuracy against KL divergence regularization to preserve inter-column dependencies. Subsequently, we train a diffusion model in this latent space, where the forward process gradually adds Gaussian noise following a linear schedule, while the reverse process learns to denoise samples through a score-based SDE.

Our synthetic data generation initially follows a 1:1 ratio with the original dataset's predefined splits (for both retrieval and test sets). The generation process involves iterative denoising of Gaussian priors to produce latent vectors, which are decoded into tabular form using the VAE's detokenizer, applying linear inverse transformations for numerical features and Softmax sampling for categorical variables. Through rigorous post-processing, including value rounding, logical consistency checks (e.g., ensuring question-level sums never exceed scale totals), and illogical case removal.

An example of our synthetic data is shown below.

```
{
  "HAMA Q4 Score": 3,
  "HAMA Q6 Score": 3,
  "HAMA Total Score": 28,
  "GAD7 Total Score": 18,
  "PHQ9 Q1 Score": 3,
  "PHQ9 Q2 Score": 2,
  "PHQ9 Q4 Score": 2,
  "PHQ9 Q9 Score": 0,
  "PHQ9 Total Score": 14,
  "HAMD Q1 Score": 2,
  "HAMD Q3 Score": 1,
  "HAMD Q4 Score": 1,
  "HAMD Q7 Score": 2,
  "HAMD Q22 Score": 1,
  "HAMD Total Score": 28,
  "BPRS Q9 Score": 3,
  "PSQI Total Score": 11,
  "SHAPS Total Score": 29,
  "HCL32 Total Score": 18,
  "DAS Total Score": 122,
  "SSRS Total Score": 40,
  "MDQ Total Score": 11,
  "BPRS Total Score": 34,
  "YMRS Total Score": 5,
  "Mood Disorder": 1
}
```

## D.2 Synthetic Data Evaluation

Building on the evaluation framework of Zhang et al. [20], we conduct a comprehensive evaluation of our synthetic data across five critical dimensions: statistical density, data quality, machine learning efficacy, privacy preservation, and logistic detectability. The results demonstrate that our synthetic dataset achieves exceptional fidelity, accurately reproducing both the core statistical patterns and complex relationships present in the original data while maintaining strong utility for machine learning applications. Notably, our evaluation shows the synthetic data achieves: (1) high-density preservation of univariate and multivariate distributions, (2) robust performance in downstream ML tasks, and (3) near-indistinguishability from real data according to rigorous detection tests. These collective findings position our synthetic data as a reliable surrogate for the original dataset across analytical and modeling use cases.

### D.2.1 Density Evaluation

The quality of synthetic data hinges on its ability to accurately replicate both individual feature distributions and the complex relationships between variables in real-world data. The metrics, derived from Shape, Trend, Coverage, and overall Density score, collectively measure the fidelity and utility of synthetic data for downstream applications.

Our synthetic dataset demonstrates remarkable fidelity in this regard, achieving an excellent overall density score of 0.86, well above the 0.8 threshold considered indicative of high-quality synthetic data. The component scores reveal particularly strong performance in capturing feature relationships, with Column Pair Trends reaching 0.93, while maintaining solid 0.79 fidelity in individual Column Shapes. These results demonstrate that our synthesis process successfully preserves the nuanced statistical patterns of the original data, with especially robust preservation of the critical multivariate relationships that are often most challenging to replicate.

Metric descriptions and our detailed scores are illustrated in the following parts.

**Shape (Column Distribution Shape).** The Shape metric evaluates how well synthetic data replicates individual feature distributions from the real data using two complementary measures: KSComplement compares cumulative distribution functions to assess overall shape alignment (including normality and skewness), while TVComplement evaluates precise probability mass matching for categorical data. Higher scores (closer to 1.0) indicate better preservation of the original data's statistical properties. As shown in Table 14, our synthetic data demonstrates strong distributional fidelity across all columns.

Table 14: Score of column distribution. KS is short for Kolmogorov-Smirnov complement; TV is short for Total Variation complement.

| Column | 0 | 1 | 2 | 3 | 4 | 5 | 6 | 7 | 8 | 9 | 10 | 11 | 12 |
|---|---|---|---|---|---|---|---|---|---|---|---|---|---|
| Metric | KS | KS | KS | KS | KS | KS | KS | KS | KS | KS | KS | KS | KS |
| Score | 0.76 | 0.74 | 0.68 | 0.79 | 0.79 | 0.84 | 0.81 | 0.85 | 0.72 | 0.77 | 0.80 | 0.80 | 0.78 |

| Column | 13 | 14 | 15 | 16 | 17 | 18 | 19 | 20 | 21 | 22 | 23 | 24 | Average |
|---|---|---|---|---|---|---|---|---|---|---|---|---|---|
| Metric | KS | KS | KS | KS | KS | KS | KS | KS | KS | KS | KS | TV | \ |
| Score | 0.82 | 0.67 | 0.78 | 0.88 | 0.80 | 0.90 | 0.84 | 0.87 | 0.90 | 0.67 | 0.83 | 0.73 | 0.79 |

**Trend (Column Pair Trends).** This metric evaluates how faithfully synthetic data reproduces the statistical relationships between variables. For numerical features (0-23), we assess linear and nonlinear correlations (CorrelationSimilarity), examining whether directional patterns (e.g., positive/negative associations) are preserved. For categorical feature 24, we measure dependency preservation through joint probability distributions (ContingencySimilarity). High scores (closer to 1.0) indicate the synthetic data successfully maintains the original data's multivariate patterns, as demonstrated in Figure 7.

**Coverage.** The Coverage metric evaluates how comprehensively synthetic data captures the full spectrum of variability in real data. For numerical features, RangeCoverage assesses whether extreme values and distribution tails are properly reproduced, while CategoryCoverage verifies the faithful representation of categorical frequencies and combinations. High scores (closer to 1.0) indicate the synthetic data successfully encompasses the complete range of real-world scenarios present in the original dataset, as shown in Table 15.

Table 15: Score of column distribution. RC is short for RangeCoverage; CC is short for CategoryCoverage.

| Column | 0 | 1 | 2 | 3 | 4 | 5 | 6 | 7 | 8 | 9 | 10 | 11 | 12 |
|---|---|---|---|---|---|---|---|---|---|---|---|---|---|
| Metric | RC | RC | RC | RC | RC | RC | RC | RC | RC | RC | RC | RC | RC |
| Score | 1.00 | 1.00 | 0.98 | 1.00 | 1.00 | 1.00 | 1.00 | 1.00 | 0.96 | 0.75 | 1.00 | 1.00 | 0.75 |

| Column | 13 | 14 | 15 | 16 | 17 | 18 | 19 | 20 | 21 | 22 | 23 | 24 |
|---|---|---|---|---|---|---|---|---|---|---|---|---|
| Metric | RC | RC | RC | RC | RC | RC | RC | RC | RC | RC | RC | CC |
| Score | 1.00 | 0.78 | 0.83 | 0.95 | 0.95 | 0.94 | 0.98 | 1.00 | 0.86 | 0.90 | 0.90 | 1.00 |

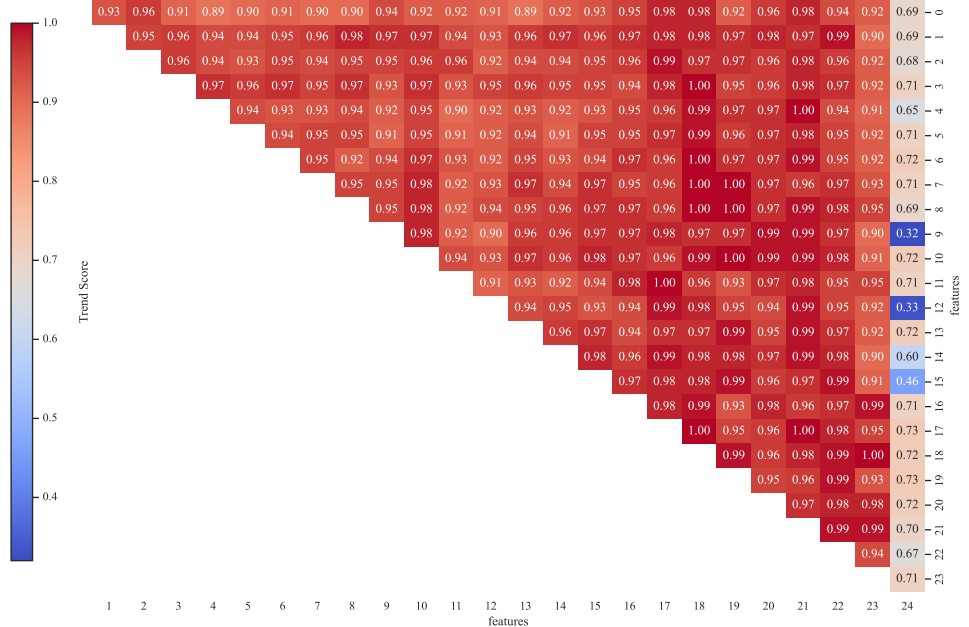

Figure 7: Column pair trend similarity scores. Features 0–23 are evaluated using CorrelationSimilarity, while feature 24 is assessed with ContingencySimilarity.

**Density (Overall Data Fidelity)** The Density score provides a comprehensive evaluation of synthetic data quality by combining distributional accuracy (Shape) and relational preservation (Trend) into a unified metric. Calculated as the harmonic mean of these components, our synthetic data scored 0.86, demonstrating exceptional statistical alignment with the source data, not only maintaining faithful individual feature distributions but also accurately reproducing the complex web of relationships between variables.

### D.2.2 Quality Evaluation

**Automatic Evaluation** We evaluate synthetic data fidelity using the Alpha-Precision metric, which assesses two critical dimensions of data quality:

Pattern Accuracy ($\alpha$=0.72): Measures how precisely the synthetic data reproduces the core statistical patterns of real data, ensuring generated samples are realistic and representative.

Diversity Coverage ($\beta$=0.44): Evaluates the synthetic data's ability to capture the full variability of real-world scenarios, including less frequent but important edge cases.

These results indicate our synthetic data successfully captures the main data distribution ($\alpha$) while showing room for improvement in representing rare cases ($\beta$). The strong $\alpha$-score suggests excellent utility for applications requiring faithful pattern reproduction, while the $\beta$-score highlights opportunities to enhance coverage of the data manifold's full extent.

**Clincian Audit** We randomly sampled 50 cases (25 real and 25 synthetic) and asked three licensed clinicians to distinguish between them. Their classification accuracy was as follows:

Table 16: Human Evaluation of MoodSyn.

| Clinician | Correct Real | Real Misclassified | Correct Synthetic | Synthetic Misclassified |
|---|---|---|---|---|
| 1 | 18 | 7 | 20 | 5 |
| 2 | 21 | 4 | 22 | 3 |
| 3 | 19 | 6 | 18 | 7 |

The three clinicians achieved accuracy rates of 0.80, 0.82, and 0.72 when classifying synthetic data as real, demonstrating the synthetic data's convincing resemblance to real clinical data. We measured

inter-annotator agreement using Fleiss' Kappa, which showed moderate agreement among raters ($\kappa$ = 0.54).

### D.2.3 Machine Learning Efficiency Evaluation

This evaluation assesses the practical utility of synthetic data by measuring how effectively it can train machine learning models to perform real-world classification tasks. We employ a comprehensive set of metrics to evaluate model performance when trained on synthetic data and tested on real data: the F1 score (both standard and weighted) evaluates the balance between precision and recall, AUROC measures the model's discriminative ability across all classification thresholds, and accuracy assesses overall prediction correctness.

As shown in Table 17, the results demonstrate our synthetic data's strong capability to train performant models, with consistently high scores across all metrics indicating successful knowledge transfer from synthetic to real data scenarios. These findings validate that models trained on our synthetic data can effectively generalize to real-world applications while maintaining robust classification performance.

Table 17: Machine Learning Efficiency (MLE) evaluation using XGBoost classifier.

| Metric | binary f1 | roc auc | weighted f1 | accuracy |
|---|---|---|---|---|
| best f1 scores | 0.884 | 0.9767 | 0.9037 | 0.9247 |
| best weighted scores | 0.884 | 0.9767 | 0.9037 | 0.9247 |
| best auroc scores | 0.8852 | 0.9735 | 0.9044 | 0.9247 |
| best acc scores | 0.884 | 0.9767 | 0.9037 | 0.9247 |
| best avg scores | 0.884 | 0.9767 | 0.9037 | 0.9247 |

### D.2.4 Privacy Evaluation

We evaluate privacy protection using the Distance to Closest Record (DCR) metric by first preprocessing data: numerical features are normalized by their range, and categorical features are one-hot encoded to standardize feature scales for consistent distance calculations. For each synthetic record, the L1 distance (sum of absolute differences) to all real records and all test records is computed, and the minimum distance for each synthetic record in both groups is identified. The DCR score is then calculated as the proportion of synthetic records where the closest real record is closer than the closest test record. A score approaching 0.5 indicates that synthetic data is equidistant to real and test datasets on average, suggesting stronger privacy protection by reducing the risk of re-identifying specific real individuals from the synthetic data.

Given that the real data has been desensitized, although we scored 0.87 for the synthetic data, it doesn't have the potential for privacy leaking. The synthetic data effectively preserves the distributional characteristics of the desensitized real data, such as the value ranges of numerical features and the probability distributions of categorical features. This is advantageous for data applications like model training, as it indicates that the synthetic data has strong "realism" or the ability to mimic the desensitized real data closely. This alignment ensures that the synthetic data retains the statistical utility of the original dataset while avoiding privacy risks associated with identifiable information, making it a valuable substitute for downstream analytical tasks.

### D.2.5 Logistic Detection Evaluation

The Logistic Detection metric evaluates how well synthetic data mimics real data by measuring its distinguishability. A perfect synthetic dataset should be virtually indistinguishable from real data.

Our synthetic data achieved an excellent score of 0.53, very close to the ideal 0.5 (random guessing). This demonstrates that our synthetic data maintains high fidelity to the statistical properties of real data, making it highly suitable for downstream applications where data realism is crucial.

# E  Experiment

## E.1  Experimental Settings

**Baseline Prompts.** For prompts in bare LLMs, we employ the key element combination format derived from medical records (Step 3 in Figure 3) along with the overall performance metrics (total score and corresponding descriptions) of each available scale. It is important to note that some scales do not rely on a total score to assess performance; instead, they use individual questions or specific question combinations to evaluate a particular dimension (e.g., CTQ scale and MCCB tests). Additionally, certain scales include not only multiple-choice questions but also open-ended questions, making it impossible to calculate a total score (e.g., NSSI scale). However, since several scales closely related to mood disorders utilize a fully multiple-choice format with a total score calculation, the information provided to the baselines remains comprehensive. A prompt template with an example is shown in Figure 8.

---

**Baseline Prompt Template**

Based on the following information, assess whether the client may have a mood disorder, which includes depression and bipolar disorder, characterized by depressive or manic symptoms. Please provide a detailed analysis in JSON format:
{"Diagnosis": yes or no, "Reasons": summarize and explain each point}
The client's medical record is as follows:
*{Medical Record (the format in Step 3)}*
The client's scale test results are as follows:
- Dysfunctional Attitudes Scale (DAS): 159/280, indicating cognitive distortions in his views of people and situations.
- Generalized Anxiety Disorder-7 (GAD-7): 2/21, indicating no anxiety symptoms.
- Hypomania Checklist (HCL-32): 11/32, indicating no hypomanic symptoms.
- Mood Disorder Questionnaire (MDQ): 7/13, suggesting possible bipolar disorder.
- Patient Health Questionnaire-9 (PHQ-9): 13/27, suggesting moderate depression.
- Snaith-Hamilton Pleasure Scale (SHAPS): 40/56, indicating anhedonia.
- Brief Psychiatric Rating Scale (BPRS): 24/126, indicating no psychotic symptoms.
- Hamilton Anxiety Rating Scale (HAMA): 3/56, indicating no anxiety.
- Hamilton Depression Rating Scale (HAMD): 5/76, indicating no depressive symptoms.
- Young Mania Rating Scale (YMRS): 11/60, indicating mild manic symptoms.

---

Figure 8: An example of baseline prompt.

**Retrieval number.** Table 18 evaluates how retrieval number $k$ affects model performance, and $k = 5$ yields the optimal balance across evaluation metrics. Note that the optimal value of $k$ may vary across datasets and should be adjusted accordingly when using the MoodAngels diagnostic framework.

Table 18: Performances with different retrieval numbers.

| Setting | ACC | MCC |
|---|---|---|
| Angel.D$_{k=5}$ | 0.923 | 0.837 |
| Angel.D$_{k=3}$ | 0.906 | 0.794 |
| Angel.D$_{k=1}$ | 0.923 | 0.733 |
| Angel.R(k=0) | 0.920 | 0.829 |

## E.2 Experimental Results

### E.2.1 Human Evaluation of Generated Reasons

The debate agents generate reasons for their position by points. We evaluated their performances by three licensed psychiatry experts on our co-author clinician's team. Metrics are clinical coherence (0-2 points), information completeness (0-2 points), and clarity of expression (0-2 points), as detailed in Table 19. The assessment results are shown in Table 20.

Table 19: Evaluation criteria of generated reasons.

| Clinical Coherence: whether reasoning aligns with clinical logic and evidence. |
| --- |
| 2 = Fully coherent and professionally sound; |
| 1 = Minor logical flaws; |
| 0 = Major errors or conflicts. |
| Information Completeness: inclusion of key diagnostic elements (e.g., symptoms, scales, course, exclusions, impairment). |
| 2 = Comprehensive and sufficient; |
| 1 = Missing important elements; |
| 0 = Lacks key diagnostic info. |
| Clarity and Expression: clarity and professionalism of the explanation. |
| 2 = Clear and professionally worded; |
| 1 = Some ambiguity or terminology misuse; |
| 0 = Unclear or confusing. |

Table 20: Evaluation results of generated reasons.

| Model | coh1 | com1 | clarity1 | coh2 | com2 | clarity2 | coh3 | com3 | clarity3 | avg_coh | avg_com | avg_clarity |
| --- | --- | --- | --- | --- | --- | --- | --- | --- | --- | --- | --- | --- |
| GPT-4o | 0.5 | 0.9 | 0.7 | 0.2 | 0.5 | 0.9 | 0.6 | 1 | 1.1 | 0.43 | 0.5 | 0.9 |
| Angel.R | 0.9 | 1.1 | 0.9 | 0.6 | 0.7 | 1.2 | 1 | 1.3 | 1.3 | 0.87 | 1.03 | 1.17 |
| Angel.D | 0.8 | 1.2 | 0.9 | 0.6 | 0.6 | 1 | 0.9 | 1.3 | 1.2 | 0.83 | 1.03 | 0.73 |
| Angel.C | 0.9 | 1.2 | 1 | 0.7 | 0.8 | 1.1 | 1.1 | 1.4 | 1.3 | 1.27 | 1.4 | 1.13 |
| multi-Angels | 1.1 | 1.3 | 1.1 | 0.8 | 0.8 | 1.1 | 1.1 | 1.1 | 1.1 | 1.3 | 1.3 | 1.1 |

We report Fleiss' Kappa as the measure of inter-annotator agreement (IAA), and the results are shown in Table 21. The results demonstrate the high quality of our generated outputs. The higher inter-annotator agreement suggests that the outputs of our multi-agent system are more consistent, clinically aligned, and better recognized by experts.

Table 21: Evaluation IAA of generated reasons.

| Model | coherence | completeness | clarity |
| --- | --- | --- | --- |
| GPT-4o | 0.6 | 0.26 | 0.13 |
| Angel.R | 0.55 | 0.37 | 0.19 |
| Angel.D | 0.69 | 0.28 | 0.25 |
| Angel.C | 0.69 | 0.33 | 0.27 |
| multi-Angels | 0.67 | 0.45 | 0.54 |

### E.2.2 Subgroup Performance Analysis

We evaluate the performance of Angel.R across demographic subgroups on the patient subset, containing 190 cases of mood disorder and 56 cases of other diseases. The performances are shown in Table 22.

For gender subgroups, despite a nominal accuracy difference ($\Delta = 0.088$), the p-value ($0.083 > 0.05$) and a comparable MCC difference ($\Delta = 0.018$) suggest no statistically significant gender bias. The slightly higher female accuracy likely reflects dataset characteristics rather than model bias.

For age subgroups, no significant accuracy difference (p=0.110) is observed between adults and children, with MCC values differing by only 0.048. It indicates that our model generalizes well across age groups without age-specific calibration.

Table 22: Subgroup Performances.

| Group | ACC | MCC |
|---|---|---|
| All Patients | 0.845 | 0.489 |
| *Gender Subgroups* | | |
| Male | 0.784 | 0.466 |
| Female | 0.872 | 0.484 |
| *Age Subgroups* | | |
| Adult | 0.795 | 0.500 |
| Female | 0.873 | 0.452 |

### E.3 Ablation Study

In this ablation study, we evaluate the robustness of our framework by changing input order and temperature.

**Input Order Sensitivity.** We use the `random.shuffle()` function to randomize the input order during the debate stage, and the results are shown in Table 23.

Table 23: Performances with different input orders.

| Setting | ACC | MCC |
|---|---|---|
| multi-Angels (ordered input) | 0.925 | 0.834 |
| multi-Angels (shuffled input) | 0.927 | 0.838 |

**Temperature Sensitivity.** We vary the temperature of GPT-4o to assess the output stability of Angel.R, and the results are shown in Table 24.

Table 24: Performances with different LLM temperatures.

| Setting | ACC | MCC |
|---|---|---|
| Angel.R$_{temp=1}$(default) | 0.920 | 0.829 |
| Angel.R$_{temp=0.7}$ | 0.922 | 0.836 |
| Angel.R$_{temp=0.3}$ | 0.923 | 0.836 |
| Angel.R$_{temp=0}$ | 0.911 | 0.808 |

These results indicate that variations in input order and temperature have minimal impact on diagnostic performance, demonstrating the stability and robustness of our framework.

### E.4 Case Study

For the inter-scale conflict and borderline cases, we selected two representative scale-only cases. The scale performance data are derived from real cases at our corporate hospital. These cases are illustrated in Figure 9 and Figure 10.

For the overlapping symptoms cases presented in this paper, which include medical records, we have anonymized all event details while retaining the patient's actual symptoms. The case is depicted in Figure 11. In our experiments, we exclusively used fully anonymized real-world cases to evaluate MoodAngels' diagnostic capabilities in practical application scenarios.

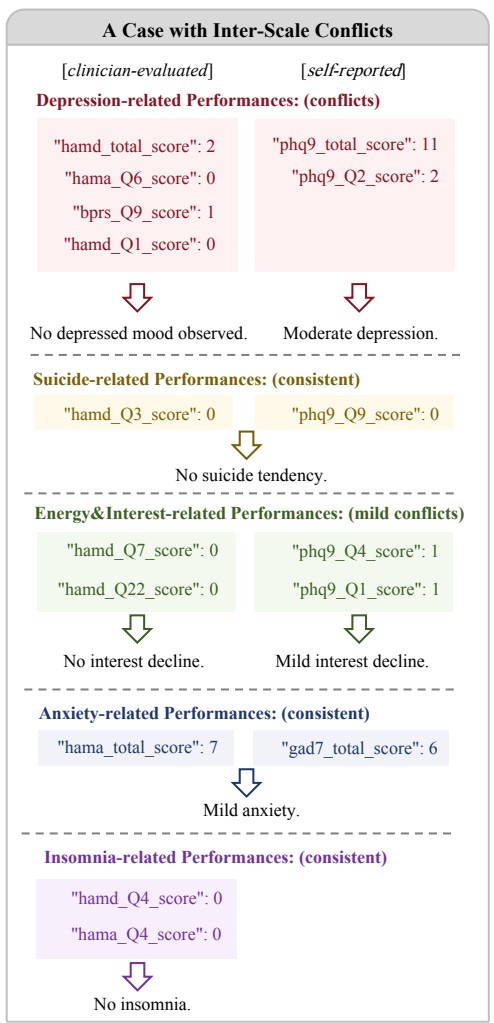

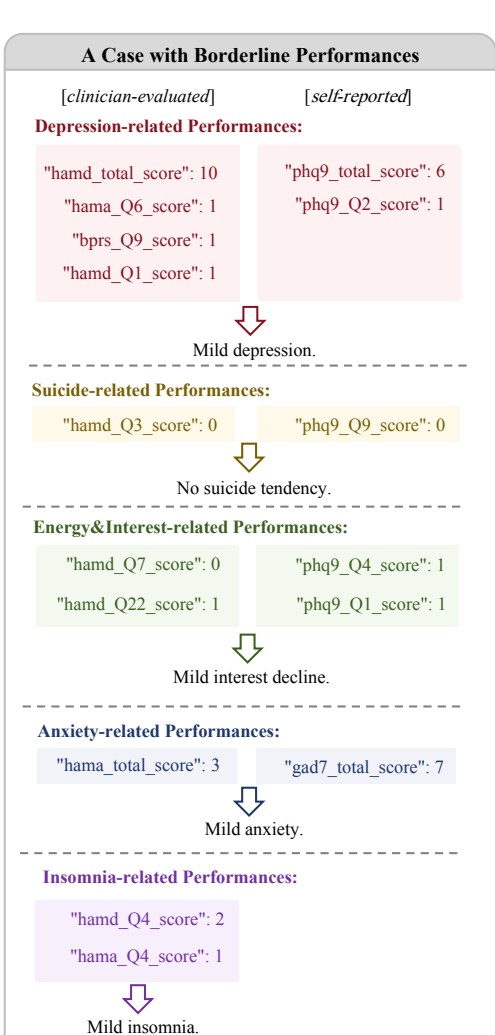

Figure 9: A case with inter-scale conflicts, with only scale performances provided. In this case, the client doesn't have any mental diseases.

Figure 10: A case with borderline performances, with only scale performances provided. In this case, the client doesn't have any mental diseases.

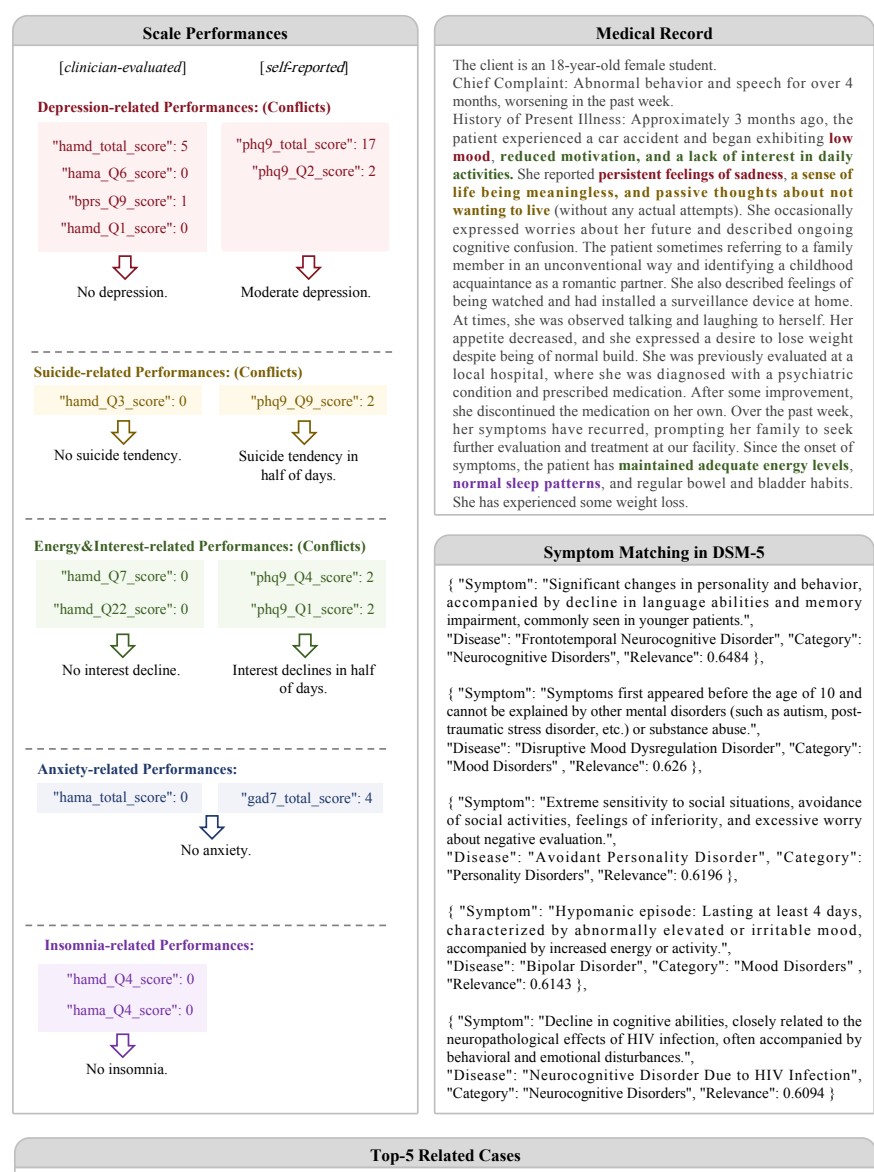

Figure 11: An adapted case with overlapping symptoms. The top-5 related cases are simplified for intuitive presentations. In this case, the client has a delusional disorder on the schizophrenia spectrum (not a mood disorder).

