# OpenReview forum: "MoodAngels: A Retrieval-augmented Multi-agent Framework for Psychiatry Diagnosis"
_NeurIPS.cc/2025/Conference — NeurIPS 2025 poster_

### Official Review · Reviewer_jxzA · 2025-06-10

**Clarity:** 3
**Significance:** 4
**Originality:** 3
**Rating:** 5
**Confidence:** 4

**Summary:**

This paper introduces MoodAngels, a retrieval-augmented multi-agent framework designed for mood disorder diagnosis. The framework addresses challenges like the subjective nature of assessments and symptom overlap. Key components of MoodAngels include granular-scale analysis of clinical assessment scales, a retrieval datastore incorporating DSM-5 diagnostic criteria and anonymized real-world clinical cases, and a suite of specialized diagnostic agents with varying reliance on historical data, culminating in a multi-agent system that uses a debate mechanism among agents and a judge agent for final diagnosis. The other contribution is the development of a open-source synthetic dataset, MoodSyn, which includes 1173 synthetic psychiatric cases for psychiatric AI research. The authors evaluate their method on real-world clinical data and MoodSyn, comparing it against several baseline LLMs. Results indicate that MoodAngels outperforms baseline LLMs in diagnostic accuracy on the real-world data.

**Questions:**

* The improvement from Angel.R to the full multi-Angels system is modest (slight improvement in GPT-4o based agents, and no better in Deepseek-V3 based agents). Meanwhile Angel.R outperforms the much more complicated variant Angel.C on real data. Does the added complexity justify this small gain (or no gain), or are there some possible qualitative benefits (e.g., robustness on specific case types) not captured by aggregate metrics? Would be nice to discuss it a bit.
* Were any statistical significance tests performed for the results in Table 3 and Table 4? If so, what were the findings?
* For the similar cases retrieval, why are the top-5 most relevant cases retrieved for Angel.D and Angel.C? Is this number fixed and how sensitive is performance to this number?
* The BGE-M3 embedder is used for symptom matching and similar case retrieval. Was its effectiveness compared to other embedding models for the psychiatric text?
* If there's no consensus after a certain number of rounds in the multi-agent debate, how does the Judge Agent make a final decision? Does such case mean the diagnosis is unspecified or uncertain?
* How was the quality and consistency of these GPT-4o generated DSM-5 knowledge ensured and evaluated?
* The real-world dataset has 56 mood disorder cases in the test set, what is the diagnosis distribution for these cases?
* In the ablation analysis, the unstructured medical records for both symptom matching and return to agent yielded the best results. Why the authors also mentioned structuring EHR is “an optimal step of reorganizing records into a structured format for precise symptom” in Appendix E.1?
* Some mood disorders can’t be clearly distinguished from each other. How did you deal with such cases? E.g. bipolar vs. MDD, bipolar vs. ADHD, or schizoaffective disorder vs. bipolar with some psychotic features.
* Note that usually we don’t use visitor/client but patient for the person who visits the clinic.

**Ethical Concerns:**

["NO or VERY MINOR ethics concerns only"]

**Final Justification:**

Why not higher
* The central claim about the superiority of the complex multi-agent debate framework is marginal from empirical evidence.
* Tasking the binary classification (disorder present/absent) rather than differentiating between specific mood disorders (e.g., Bipolar I vs. MDD) makes the task less complex but also reduces the perceived significance from the clinical use perspective.

Why not lower
* Angel.R model itself is a novel and effective system that significantly outperforms strong LLM baselines on a challenging real-world dataset.
* Open-source MoodSyn dataset is a significant service to the psychiatric AI research community.
* The methodology and rebuttal are well-detailed with its capabilities and limitations.

**Limitations:**

Limitations are mentioned and discussed in the appendix. The potential negative societal impact has not been explicitly discussed in this work. It would be nice to discuss the societal impact, specifically the caveats and warnings of using AI for psychiatric diagnosis. Meanwhile it's also good to point out what aspects of psychiatric diagnosis may not be captured well with such proposed method. e.g., if having subgroup analysis for each type of mood disorder, the authors may point out what types of mood disorder are suitable for using the method, what what types are not.

**Paper Formatting Concerns:**

No paper formatting concerns.

**Quality:**

3

**Strengths And Weaknesses:**

Strengths
* The authors’ proposed system, MoodAngel, is a tailored multi-agent system designed specifically for the complexities of psychiatric diagnosis, incorporating granular data analysis, knowledge retrieval, and debate reasoning.
* The paper introduces MoodSyn, a synthetic dataset for mood disorders, along with a detailed account of its generation and a rigorous multi-faceted evaluation. This is a valuable resource for the community.
* The framework integrates multiple important components: item-level scale analysis, DSM-5 and case-based retrieval, and a multi-agent debate architecture. The data processing steps, although in the Appendix, are also detailed and thoughtful.
* The MoodAngels agents, particularly Angel.R, demonstrate a substantial improvement on real-world clinical cases.
* The paper is supported by detailed appendices covering ethical considerations, related work, data processing, synthetic data evaluation, and agent details, which greatly enhance the transparency and depth of the work.
* AI for psychiatry is very challenging due to data subjectivity, privacy, and lack of objective markers. This work makes a concerted effort to tackle these issues.

Weaknesses
* While Angel.R shows significant uplift over baselines, the incremental performance gain from Angel.R to the full multi-Angels system (with debate) is relatively small. This raises questions about the cost-benefit of the added complexity. Specifically the paper does not discuss why the proposed multi-agent framework do not show a clearer advantage compared with the simple agent setup, or in some cases perform slightly worse. It would be better to add some discussion about this.
* The experimental results in Tables 3 and 4 are presented without statistics such as confidence intervals or p-values. Given some small differences between agent variants, statistical tests would be necessary to confirm the significance of these differences.
* While the ablation study examines medical record format and scale selection for Angel.R, an ablation on the impact of the retrieval component (e.g., w/o DSM-5, w/o similar cases) or the debate mechanism (e.g., no debate, just majority vote among agents) itself would be beneficial.
* The paper mentions many default parameters for the study. The impact of choices like the number of retrieved cases (top-5), or thresholds in granular-scale analysis (top-5% correlation), is not discussed.

---

> ### Author Rebuttal · Authors · 2025-07-31
>
> We really acknowledge your thoughtful questions, and we revised our paper to clarify them and add experiments on the ablation study and analysis.
>
> **W1/Q1 Clinical Significance of Incremental Performance Gains**
> While the improvement from Angel.R to the multi-agent system may appear modest in statistical terms, even marginal gains in diagnostic accuracy hold substantial clinical value. In psychiatric practice, where each case carries significant implications for patient care and treatment outcomes, these incremental advancements can translate into meaningful benefits for a broader population of individuals seeking mental health support.
>
> **W2/Q2 Statistical Significance Test Reults**
> - Interpretation of p-values: All models except llama3-8b-instruct on real cases have p-values far less than 0.05, indicating that their predictive performance is significantly better than random guessing.
>
> - 95\% Confidence intervals (CI): The 95% confidence intervals for each model are relatively narrow, suggesting that the results have high statistical reliability.
>
> Results of real cases:
> | Model | ACC |95\% CI | p value |
> |:-------|:-----:|-----:|-----:|
> | llama3-8b-instruct | 0.506 | [0.464, 0.548] | 0.80 |
> | mistral7b-instruct-v0.3 | 0.597 | [0.555, 0.638] | 0.00 |
> | MedGemma-27b | 0.704 | [0.664, 0.742] | 1.58e-22 |
> | gpt-4o | 0.797 | [0.761, 0.829] | 1.46e-47 |
> | deepseek-v3 | 0.847 | [0.814, 0.876] | 3.55e-66 |
> | [gpt]Angel.R | 0.920 | [0.894, 0.941] | 2.00e-102 |
> | [gpt]Angel.D | 0.923 | [0.898, 0.944] | 1.47e-104 |
> | [gpt]Angel.C | 0.914 | [0.888, 0.936] | 2.65e-99 |
> | [gpt]multi-angels | 0.925 | [0.900, 0.946] | 1.22e-105 |
> | [deepseek]Angel.R | 0.927 | [0.902, 0.947] | 9.80e-107 |
> | [deepseek]Angel.D | 0.920 | [0.894, 0.941] | 2.00e-102 |
> | [deepseek]Angel.C | 0.922 | [0.896, 0.942] | 1.74e-103 |
> | [deepseek]multi-angels | 0.923 | [0.898, 0.944] | 1.47e-104 |
>
> Results of synthetic cases:
> | Model | ACC |95\% CI | p value |
> |:-------|:-----:|-----:|-----:|
> | llama3-8b-instruct | 0.564 | [0.478, 0.648] | 0.15 |
> | mistral7b-instruct-v0.3 | 0.636 | [0.550, 0.715] | 0.00 |
> | MedGemma-27b | 0.743 | [0.662, 0.813] | 7.60e-09 |
> | deepseek-v3 | 0.821 | [0.748, 0.881] | 5.41e-15 |
> | [deepseek]Angel.R | 0.800 | [0.724, 0.863] | 4.25e-13 |
> | [deepseek]Angel.D | 0.807 | [0.732, 0.869] | 1.04e-13 |
> | [deepseek]Angel.C | 0.821 | [0.748, 0.881] | 5.41e-15 |
> | [deepseek]multi-angels | 0.821 | [0.748, 0.881] | 5.41e-15 |
>
> **W3. Ablation Study on Retrieval Components** The strong performance of majority voting validates the diagnostic capability of individual agent judgments. However, the debate process remains essential for ensuring comprehensive symptom analysis through structured deliberation, maintaining output stability via evidence-based adjudication, and providing interpretable reasoning traces for clinical validation.
>
> | Setting| ACC | |MCC | |
> |:-------|:-----:|-----:|-----:|-----:|
> Angel.R|0.927||0.841||
> Angel.R w/o DSM5 Symptom Matching|0.923|0.004↓|0.83|0.011↓|
>
> | Setting|ACC|| MCC||
> |:-------|:-----:|-----:|-----:|:-------|
> multi-angels debate |0.925 ||0.834||
> multi-angels majority vote| 0.923|0.002↓|0.835|0.001↑|
>
> **W4. Top-5% Scale-Item Threshold Selection**
> In designing our threshold criteria, we carefully considered this important aspect. Through a review of established psychiatric methodologies [1], we observed that prior work typically derives thresholds through a combination of clinical expertise and empirical data distribution analysis. To ensure clinical validity, we first computed Pearson correlation coefficients for each scale item, then engaged in extensive discussions with our clinician co-authors to determine appropriate thresholds. Notably, based on their expert recommendations, we incorporated additional items (PHQ-9 Q1 and Q2) that demonstrated particular clinical relevance, as detailed in lines 697-700 of our manuscript.
>
> [1]Unraveling the distinction between depression and anxiety: A machine learning exploration of causal relationships.
>
> **W4/Q3. Retrieval Top-k Similar Cases Threshold Selection**
> We will include an ablation study that varies the top-k value (0, 1, 3, 5) for case retrieval, and we will report the results in the rebuttal once testing is complete. Notably, we observed that Angel.R already performs well even without retrieval (k=0), which further supports the robustness and quality of our overall framework.
>
> **Q4. Choice of Retriever** We chose BGE-M3 for its strong semantic representation capabilities, which align well with our goal of capturing nuanced symptom similarities in clinical texts. While we agree that exploring alternative retrievers could offer additional insights, our current focus has been on validating the overall effectiveness of the framework. Notably, as shown in W3, even in the absence of any retrieval component (Angel.R w/o DSM-5 Symptom Matching), the model still achieves promising results, suggesting the robustness of the reasoning process. We hope this addresses your concern, and we truly value your encouraging feedback.
>
> **Q5. Judge's Criteria for Ending the Debate.** We designed the prompt of the Judge Agent to simulate a debate format, with a minimum of 2 rounds and a maximum of 4. At the beginning of each round, the prompt guides the Pro and Con agents through a structured process: Round 1 for opening statements, Round 2 for rebuttals, Round 3 for clarifying unresolved points, and the final round for closing arguments. After each round, the Judge Agent evaluates whether the debate should continue, provides reasoning, and reflects on the arguments. At the end of the debate, the judge delivers a diagnosis with reasons.
>
> **Q6. Quality of Knowledge Base.** The diagnostic and differential criteria from the DSM-5 were originally presented as bullet points, as shown in Appendix C1. We used GPT-4o to revise these items into concise, self-contained statements, rather than generating new diagnostic content. This step was necessary because some original entries contained pronouns or overly detailed explanations that were not helpful for symptom matching between the knowledge base and medical records. After constructing the knowledge base, each extracted item was reviewed and verified by our domain expert.
>
> **Q7. Distribution of real cases** We are calculating the detailed types, and we will report the result in this rebuttal.
>
> **Q8. Medical Record Structuring** We designed medical record structuring as an optional component to ensure the flexibility of our framework, particularly in scenarios where medical records lack clarity or where input token limitations need to be considered.
>
> **Q9. Distinguishing Mood Disorders.** Currently, our framework performs binary classification—determining whether a mood disorder is present. While we have extracted the DSM-5 differential criteria for more fine-grained distinctions, we do not attempt subcategory classification. This is because psychiatric diagnosis requires careful deliberation; in clinical practice, clinicians typically first identify a broad diagnostic direction before narrowing it down through further assessments. Similarly, our framework is designed to support this initial diagnostic stage.
>
> **Q10. Terminology for Individuals Seeking Help** Thanks for raising this important point. We also place great importance on using professionally appropriate terminology. For individuals who have not been formally diagnosed with a mental disorder, we use terms such as client or visitor to show respect and avoid assumptions. Once a diagnosis is confirmed, we refer to them more specifically as patients. For example, as in line 276, where we state “a patient with severe depression.”
>
> **L1. Usage Declaration of the Diagnostic Framework** Thanks. We will revise our paper and GitHub repo to add a usage declaration.

---

> > ### Author Response · Authors · 2025-08-03
> > **Additional Results**
> >
> > **Q3**  In this ablation study, we evaluate how the number of retrieved similar cases (k) affects model performance. Experimental results demonstrate that k=5 yields the optimal balance across evaluation metrics.
> > Setting|ACC|MCC|
> > |:----:|----:|----:|
> > Angel.D$_{k=5}$|0.923|0.837
> > Angel.D$_{k=3}$|0.906|0.794
> > Angel.D$_{k=1}$|0.923|0.733
> > Angel.R($k=0$)|0.920|0.829
> >
> > **Q7** Thank you for your interest in the detailed subtype distribution of mood disorder cases in our test set. Among the 190 mood disorder cases, the diagnostic subtypes are distributed as follows:
> > - Bipolar I Disorder, current episode manic: 42 cases
> > - Bipolar I Disorder, current episode major depressive: 23 cases
> > - Bipolar II Disorder, current episode depressive: 50 cases
> > - Bipolar II Disorder, current episode hypomanic: 2 cases
> > - Major Depressive Disorder: 55 cases
> > - Comorbidity with other disorders: 12 cases
> > - Bipolar I Disorder, current episode mixed: 2 cases
> > - Bipolar I Disorder, cyclothymic episode: 1 case
> > - Bipolar I Disorder, in remission: 2 cases
> > - Bipolar II Disorder, mixed episode: 1 case

---

> ### Comment · Area_Chair_weSh · 2025-08-02
>
> Dear Reviewer,
>
> The author has provided a rebuttal to respond to your comments. Please have a read on the author response and discuss with author if necessary.
>
> Thanks,
>
> AC Team

---

> ### Comment · Reviewer_jxzA · 2025-08-03
> **Thanks for the comprehensive rebuttal**
>
> I appreciate the new ablation studies and clarifications to my questions. Your rebuttal have strengthened the paper (please add them into the main context or appendix if possible) and addressed most of my initial concerns.
>
> Note that your response has also raised a point for discussion that shows removing DSM-5 symptom matching or replacing the debate with a simple majority vote leads to a very minimal drop in performance. This reinforces my original concern (Q1): the quantitative evidence does not quite support the necessity of the added complexity of the full multi-agent debate system over the simpler, high-performing Angel.R or a simple majority vote. Also, while I respect your argument about the clinical value of marginal gains, the paper would be more balanced if it explicitly acknowledged that the empirical performance boost from the debate is not very significant. You may highlight, as you do in the rebuttal, that its value lies in other areas like interpretability or handling edge cases or something else. This transparency would make the manuscript more robust. Finally, your response for Q9 is also a detail that helps in scoping the contributions and limitations. I recommend making this explicit in the paper.
>
> I am raising my score from 4 to 5. The authors provided a helpful rebuttal that addressed nearly all of my concerns with ablation experiments, statistical analyses, and detailed clarifications.
>
> Why not higher/lower see final justification section.

---

> > ### Author Response · Authors · 2025-08-07
> > **Appreciation for the Reviewer's Thoughtful Feedback and Engagement**
> >
> > Dear Reviewer jxzA:
> >
> > We sincerely thank you for the thoughtful comments, constructive suggestions, and the time spent carefully engaging with our work. We truly appreciate your recognition of our efforts in addressing your concerns through additional experiments and clarifications. We will incorporate these insights into the final version to make the paper more transparent and robust.
> >
> > MoodAngels Authors

---

### Official Review · Reviewer_6o25 · 2025-07-02

**Clarity:** 2
**Significance:** 2
**Originality:** 2
**Rating:** 3
**Confidence:** 3

**Summary:**

The paper introduces a retrieval-augmented multi-agent system for psychiatric diagnosis. Specifically, it proposes a granular item-level analysis of psychiatric scales by integrating DSM-5 structured retrieval and generates a synthetic dataset (MoodSyn) to address data privacy issues. This study is validated on a hybrid dataset with both real-world cases and synthetic cases.

**Questions:**

See above.

**Ethical Concerns:**

["Major Concern: Discrimination, bias, and fairness"]

**Final Justification:**

The reviewer is still not fully convinced about the clinical realism and practical implementation of the study in clinical trials.

**Limitations:**

The study lacks clinical justification and validation.

**Quality:**

2

**Strengths And Weaknesses:**

Strengths:

- The challenge of AI-assisted mood disorder diagnosis is important.

- The reviewer acknowledges that the authors took privacy concerns into consideration.


Weaknesses:

1. Insufficient Clinical Validity of the Proposed Framework
- There is no involvement of licensed clinical professionals in validating the diagnostic outputs, aside from general references to coauthors.

- The design of the Granular-scale Analysis lacks justification regarding its clinical validity and reliability.

- Mood disorders are highly dependent on clinical interviews and patient interaction; thus, they cannot be reliably diagnosed from scale scores and written notes alone.

2. Lack of Clinical Realism in the Synthetic Dataset
- No human evaluation by licensed mental health professionals or comparison with gold-standard clinical diagnoses is provided.

- Statistical similarity $\neq$ clinical adequacy: Diagnostic decisions depend on subtle symptom patterns and context, not just distributional alignment.

- There is no evidence that the synthetic dataset improves model performance on downstream or real-world tasks.

- The dataset's clinical representativeness is unclear, and no comparisons are made to existing synthetic or anonymized psychiatric datasets.

Overall, the authors fail to evaluate the realism, practical utility, and clinical usefulness of the synthetic dataset.

3. Unfair Baseline Comparisons

- GPT-4o and other baselines are neither (1) designed for psychiatric diagnosis nor (2) fine-tuned for it, making the performance comparisons methodologically unsound and misleading.

---

> ### Author Rebuttal · Authors · 2025-07-31
>
> **W1.1 Clinical validation of diagnostic output** We evaluated their performances by three licensed psychiatry experts on our co-author clinician’s team. Metrics are clinical coherence (0-2 points),  information completeness (0-2 points), and clarity of expression (0-2 points).
>
> Clinical Coherence: whether reasoning aligns with clinical logic and evidence.
> - 2 = Fully coherent and professionally sound
> - 1 = Minor logical flaws
> - 0 = Major errors or conflicts
>
> Information Completeness: inclusion of key diagnostic elements (e.g., symptoms, scales, course, exclusions, impairment).
> - 2 = Comprehensive and sufficient
> - 1 = Missing important elements
> - 0 = Lacks key diagnostic info
>
> Clarity and Expression: clarity and professionalism of the explanation.
> - 2 = Clear and professionally worded
> - 1 = Some ambiguity or terminology misuse
> - 0 = Unclear or confusing
>
> Here are the assessment results:
> |Model|coh1|com1|clarity1|coh2|com2|clarity2|coh3|com3|clarity3|avg_coh|avg_com|avg_clarity|
> |:-------|:-----:|-----:|-----:|-----:|-----:|-----:|-----:|-----:|-----:|-----:|-----:|-----:|
> GPT-4o|0.5|0.9|0.7|0.2|0.5|0.9|0.6|1|1.1|0.43|0.5|0.9|
> Angel.R|0.9|1.1|0.9|0.6|0.7|1.2|1|1.3|1.3|0.87|1.03|1.17|
> Angel.D|0.8|1.2|0.9|0.6|0.6|1|0.9|1.3|1.2|0.83|1.03|0.73|
> Angel.C|0.9|1.2|1|0.7|0.8|1.1|1.1|1.4|1.3|1.27|1.4|1.13|
> multi-Angels|1.1|1.3|1.1|0.8|0.8|1.1|1.1|1.1|1.1|1.3|1.3|1.1|
>
> We report Fleiss' Kappa as the measure of inter-annotator agreement (IAA).
> |Model|coherence|completeness|clarity|
> |:-------|:-----:|-----:|-----:|
> GPT-4o|0.6|0.26|0.13|
> Angel.R|0.55|0.37|0.19|
> Angel.D|0.69|0.28|0.25|
> Angel.C|0.69|0.33|0.27|
> multi-Angels|0.67|0.45|0.54|
>
> The results demonstrate the high quality of our generated outputs. **The higher inter-annotator agreement suggests that the outputs of our multi-agent system are more consistent, clinically aligned, and better recognized by experts.**
>
> **W1.2 Literature Basis for Item-Level Analysis** Traditional psychiatric diagnostic approaches have predominantly relied on total score interpretations of assessment scales, potentially obscuring crucial symptom-level heterogeneity. Recent advances in depression research have highlighted the importance of granular-scale analysis to overcome this limitation. [1] demonstrated how item-level variability in depression scales (e.g., distinguishing somatic vs. cognitive symptoms) challenges conventional unidimensional scoring. Beyond this, [2] identified core depressive symptoms through factor analysis of individual Hamilton Scale items, while van [3] conclued clinically meaningful MDD subtypes using PHQ-9 item clustering. Through close collaboration with clinical experts, our MoodAngels framework extends this paradigm by systematically grouping scale items to reduce assessment bias and enhance diagnostic precision.
>
> [1] Measuring depression over time… Or not? Lack of unidimensionality and longitudinal measurement invariance in four common rating scales of depression. In *Psychological Assessment*, 28(11), 1354–1367.
>
> [2] An Item Response analysis of the Hamilton Depression Rating Scale using shared data from two pharmaceutical companies. In *Journal of Psychiatric Research*, 38(3), 275-284.
>
> [3] Data-driven subtypes of major depressive disorder: a systematic review. In *BMC medicine*, 10(1), 156.
>
> **W1.3 Dependence on Clinical Interviews** Our framework incorporates medical records and five clinician-administered scales, all of which rely on structured clinical interviews. Rather than replacing clinical interactions, our system is designed to assist clinicians after the full interview process is completed, by supporting diagnostic reasoning based on the comprehensive set of information gathered from the patient.
>
> **W2.1/W2.3/W2.4 Human Evaluation of Synthetic Dataset** Our synthetic dataset consists of scale-based numerical data. We initially filtered the data by ensuring the sum of subscores did not exceed the total score. Additionally, we evaluated how closely the synthetic data resembled real data using automated metrics (see Line 877, achieving a score of 0.87). For human evaluation, we randomly sampled 50 cases (25 real and 25 synthetic) and asked three licensed clinicians to distinguish between them. Their classification accuracy was as follows:
>
> |Clinician|Correct Real|Real Misclassified|Correct Synthetic|Synthetic Misclassified|
> |:-----|:------:|-------:|-------:|-------:|
> 1|18|7|20|5|
> 2|21|4|22|3|
> 3|19|6|18|7|
> The accuracy rates for identifying synthetic data as "real" were 0.80, 0.88, and 0.72 across the three clinicians, demonstrating that the synthetic data could convincingly mimic real data.
>
> **W2.2 Pattern Accuracy of Synthetic Data** In line 847, we evaluated Pattern Accuracy (α=0.72), which measures how precisely the synthetic data reproduces the core statistical patterns of real data, ensuring generated samples are realistic and representative.
>
> **W2.4 Comparisons to existing psychiatry datasets** Current scale data are mainly one self-reported scale for the depression detection task. We found no existing dataset enough for mood disorder diagnosis. Therefore, besides our main contribution of designing a diagnostic framework and testing its performance on real clinical data,  we constructed a synthetic dataset to prompt academic research in the field.
>
> **W3. Medical baselines** We tested the result of medgemma-27b-text-it, which is an advanced LLM provided by Google. We found that in real cases, it tends to diagnose nearly every case with mood-disorder-related symptoms in the medical record as mood disorder, without having the sensitivity of overlapping symptoms in other mental disorders.
>
> Result on synthetic data: recall: 0.67, acc: 0.743, mcc: 0.57, macro f1: 0.717.
>
> Result on real data: recall: 0.535, acc: 0.704, mcc: 0.525, macro f1: 0.704.

---

> ### Comment · Area_Chair_weSh · 2025-08-02
> **Please read author rebuttal**
>
> Dear Reviewer,
>
> The author has provided a rebuttal to respond to your comments. Please have a read on the author response and discuss with author if necessary.
>
> Thanks,
>
> AC Team

---

### Official Review · Reviewer_EzJT · 2025-07-02

**Clarity:** 4
**Significance:** 2
**Originality:** 2
**Rating:** 4
**Confidence:** 4

**Summary:**

The paper proposes a multi-agent psychiatric diagnostic framework for mood disorders that utilizes retrieval-augmented generation to incorporate the context of analogous cases. The architecture is composed of distinct agents: Angel.R, which operates without reference to previous cases; Angel.D, which presents retrieved cases as contextual information; and Angel.C, which conducts a comparative analysis between each retrieved case and the current query, returning this analysis as context. In the event of diagnostic disagreements, supplementary agents engage in positive and negative argumentation to facilitate resolution. Concurrently, the paper introduces MoodSyn, an open-source dataset comprising 1,173 synthetic cases. The framework demonstrates superior performance, surpassing that of GPT-4o by 12.3%.

**Questions:**

- Given the potential for performance improvements, are there plans to evaluate models that have been fine-tuned or explicitly designed for this clinical context?

- Was the choice to use a combination of variational autoencoders and diffusion models for data generation benchmarked against other approaches? If so, what was the rationale for selecting this specific method over other potential alternatives?

**Ethical Concerns:**

["NO or VERY MINOR ethics concerns only"]

**Final Justification:**

I have raised my score.

The clarification on the specific methodological innovations was helpful, but the new results demonstrating the limitations of current medical Large Language Models (LLMs) were particularly compelling. These findings significantly strengthen the paper's contribution and impact.

The rebuttal has convinced me that the paper is stronger than I had initially perceived. Consequently, I have raised my score to reflect this positive reassessment.

**Limitations:**

yes

**Paper Formatting Concerns:**

.

**Quality:**

3

**Strengths And Weaknesses:**

Strengths:
- The assertions made within the paper are well-substantiated, with the experimental outcomes providing robust corroboration for the claims.
- The study is presented with notable clarity and is both well-written and coherently structured.
- The authors have provided code and rich appendices, which significantly strengthen the paper's reproducibility and utility.


Weaknesses:
- The primary weakness is the limited novelty of the proposed framework. As it primarily combines existing components, the contribution may not meet the high bar for novel methods expected at NeurIPS. Given its strong application, the work might be better suited for a top-tier healthcare informatics venue (e.g., ICHI, CBMS) or a journal, where the applied contribution would be more prominently recognized.

- A discussion on the potential benefits of using specialized, fine-tuned models would be a valuable addition. Exploring this could offer a more comprehensive understanding of the framework's place within the existing landscape of NLP methods.

---

> ### Author Rebuttal · Authors · 2025-07-31
>
> **W1. Clarification on Methodological Innovations** We appreciate the reviewer’s feedback and would like to clarify that our framework is not merely a combination of existing components, but introduces several key innovations tailored to the unique challenges of psychiatric diagnosis:
> - Granular Scale Decomposition: Instead of treating scales as fixed global scores, we restructure them into item-level diagnostic signals. This mitigates the limitations of self-reported or clinician-rated scale scores by enabling finer-grained and more interpretable analysis.
> - Retrieval-Augmented Symptom Matching: Unlike generic RAG applications, we apply retrieval specifically to address the symptom overlap problem in psychiatric diagnosis, leveraging structured DSM-5 criteria and expert-annotated examples to improve diagnostic precision in ambiguous cases.
> - Debate-Inspired Multi-Agent Reasoning: We introduce a novel debate mechanism, inspired by formal debate tournaments, in which independent diagnostic agents present their reasoning and a non-intervening judge agent synthesizes the outcome. This design enables transparent multi-perspective evaluation and handles difficult or conflicting cases with human-like deliberation.
>
> Together, these innovations go beyond simple component reuse and are purpose-built for the nuanced reasoning required in mental disorder diagnosis.
>
> **W2/Q1. Medical LLM as backbone** We are testing the results of medical llm as backbone, and we will report the result in this rebuttal as soon as we finish the experiment.
>
> **Q2. Synthetic data generation methods** We appreciate the insightful suggestion regarding synthetic data. While our primary contribution lies in the diagnostic framework, the synthetic dataset was developed as an additional effort to address the challenges of data privacy and the lack of high-quality public datasets in psychiatric diagnosis. Currently, we employ an off-the-shelf method for case generation, and we are actively exploring ways to incorporate more diverse datasets and multimodal inputs to further advance AI-driven psychiatric diagnosis.

---

> > ### Author Response · Authors · 2025-08-04
> > **Medical LLMs' Performances**
> >
> > **Medical baselines** We tested the result of medgemma-27b-text-it, which is an advanced LLM provided by Google. We found that in real cases, it tends to diagnose nearly every case with mood-disorder-related symptoms in the medical record as mood disorder, without having the sensitivity of overlapping symptoms in other mental disorders. (The performance is worse than Deepseek-v3.)
> >
> > Result on synthetic data: recall: 0.67, acc: 0.743, mcc: 0.57, macro f1: 0.717.
> >
> > Result on real data: recall: 0.535, acc: 0.704, mcc: 0.525, macro f1: 0.704.
> >
> > **Medical LLMs as Agent Backbones**
> > We sincerely appreciate your thoughtful suggestion. In our experiments, however, we found that current medical LLMs are not yet sufficiently capable to serve as the backbone of autonomous diagnostic agents, particularly in tasks that require long-context reasoning, strict instruction-following, and clinical decision-making.
> >
> > This limitation appears to be a broader challenge across the field. For instance, the AgentClinic benchmark [1] from Stanford evaluates clinical agent performance using general-purpose LLMs such as GPT-4o. Similarly, AI Hospital [2] adopts GPT-4 as its backbone, and Agent Hospital [3] is built on GPT-3.5. These choices reflect a growing consensus that current domain-specific medical LLMs, while rich in knowledge, often underperform in tasks demanding sustained reasoning and decision-making autonomy.
> >
> > In our evaluation, we tested several representative medical LLMs, including MedGEMMA, HuoTuo-GPT2, and MedLLaMA. We observed that MedGEMMA failed to produce output when presented with long-context inputs; HuoTuo-GPT2 often declined to make a diagnosis, requesting more information instead; and MedLLaMA generated lengthy and repetitive analyses without reaching a clear conclusion. These behaviors highlight current limitations in instruction adherence and clinical reasoning robustness, rather than any framework-specific compatibility issues.
> >
> > We remain optimistic about the future potential of medical LLMs and look forward to integrating them as their capabilities mature.
> >
> > [1] AgentClinic: a multimodal agent benchmark to evaluate AI in simulated clinical environments.
> >
> > [2] AI Hospital: Benchmarking Large Language Models in a Multi-agent Medical Interaction Simulator.
> >
> > [3] Agent Hospital: A Simulacrum of Hospital with Evolvable Medical Agents.

---

> > ### Comment · Reviewer_EzJT · 2025-08-06
> >
> > I thank the authors for the detailed rebuttal and the new experiments. The clarification on the specific methodological innovations and, especially, the new results showing the limitations of current medical LLMs were very compelling.
> >
> > These findings have addressed my main concerns and significantly strengthened the paper. While I am still deliberating on the final recommendation, the rebuttal has convinced me that the paper is stronger than my initial assessment.

---

> ### Comment · Area_Chair_weSh · 2025-08-02
> **Please read author rebuttal**
>
> Dear Reviewer,
>
> The author has provided a rebuttal to respond to your comments. Please have a read on the author response and discuss with author if necessary.
>
> Thanks,
>
> AC Team

---

### Official Review · Reviewer_HsoZ · 2025-07-05

**Clarity:** 3
**Significance:** 4
**Originality:** 3
**Rating:** 5
**Confidence:** 3

**Summary:**

This paper presents MoodAngels, a multi-agent framework for psychiatric diagnosis, particularly for mood disorders. It combines fine-grained analysis of DSM-5 scales, a retrieval-augmented knowledge module, and a structured multi-agent debate system (Angel.R/D/C, Judge, and Debate Agent). The authors also introduce MoodSyn, a synthetic benchmark dataset built via a combination of generative techniques to simulate high-quality psychiatric cases while preserving privacy. Evaluations on real hospital data demonstrate that the system outperforms strong baselines, including GPT-4o.

**Questions:**

1. Could you provide a detailed example of a case where Angel.R, Angel.C and Angel.D disagree, and show how the Debate Agent arguments are structured and evaluated by the Judge?

2. How does the Judge determine when a debate should end? Is there a scoring or threshold-based mechanism?

3. Have you evaluated the consistency of final decisions under changes to input phrasing, order of Angel outputs, or prompt style?

4. Is the Debate Agent using scripted arguments or generating open-ended reasoning? How interpretable are the generated explanations?

5. Does the Judge module retain any clinical rules (e.g., DSM-5 heuristics), or is it entirely language-model-based?

**Ethical Concerns:**

["NO or VERY MINOR ethics concerns only"]

**Limitations:**

Partially addressed. The synthetic dataset's limits are discussed, and the authors recognize the challenge of capturing full clinical nuance. However, the paper would benefit from more detailed reflection on the debate module’s sensitivity to randomness and prompt design, as well as potential unintended bias amplification across agents.

**Paper Formatting Concerns:**

There are no major formatting issues in this paper.

**Quality:**

4

**Strengths And Weaknesses:**

This paper is timely and ambitious. It addresses core limitations in existing medical LLMs, namely: limited interpretability, lack of structure in multi-perspective reasoning, and the absence of privacy-aware psychiatric benchmarks. The design of agent specialization (retrieval, contrastive, and isolated judgment) is thoughtful and clinically grounded, reflecting patterns observed in real-world differential diagnosis.

However, the core decision aggregation mechanism—namely, the Judge and Debate Agent modules—remains underexplained from an interpretability and implementation perspective. While the paper outlines the overall pipeline (i.e., initial vote → conflict check → debate → final decision) (lines 173-181), it does not show how the Judge actually resolves conflicts or evaluates arguments. For instance, what mechanisms (prompt patterns, learned scoring functions, or rule-based criteria) are used by the Judge to choose between competing perspectives? How are the debate rounds terminated, and how sensitive is the final outcome to minor prompt variations or phrasing changes?

Moreover, no concrete case studies or dialogue traces are provided to help the reader understand what types of reasoning emerge during debates. This makes it difficult to assess whether the system is simply ensembling diverse outputs or performing meaningful deliberation. In recent literature on medical LLM auditing (e.g., checklists in MedQA, Dialogue-Based Evaluation in MedAlign), providing interpretable reasoning paths has become a standard expectation. Including such examples would significantly enhance trustworthiness and traceability.

The paper could also benefit from a more thorough evaluation of system robustness—e.g., does MoodAngels produce consistent diagnoses under slight perturbations (e.g., varying agent order, temperature, phrasing)? Multi-agent LLM systems are known to be brittle, and demonstrating stability would strengthen the case for real-world deployment.

---

> ### Author Rebuttal · Authors · 2025-07-30
>
> We are grateful for your thoughtful questions, and we have revised our paper to address them and presented examples of our framework's output.
>
> **Q1.** A debate case, where only Angel C was correct and the other two were wrong, and the Judge made a correct decision by debate.
>
> [Angel R's final output] The client has a mood disorder.
> - The client shows mood instability with shifts from depression to (hypo)mania, consistent with bipolar disorder.
> - Past hospitalization for mood disorder and long-term use of mood stabilizers support this.
> - Symptoms like irritability, increased activity, and energy align with hypomanic episodes per DSM-5.
>
> [Angel D's final output] The client has a mood disorder.
> - The client exhibits emotional instability, including hallucinations and inappropriate laughter.
> - Despite clinician-rated scales showing no depression, self-reports suggest moderate depression.
> - Similar cases were diagnosed as bipolar or major depression, reinforcing this possibility.
>
> [Angel C's final output] The client doesn't have a mood disorder.
> - Though the client self-reports mild depressive and anxiety symptoms, clinician ratings show no signs of depression, anxiety, or insomnia.
> - Mood fluctuation history is noted, but current assessments do not support an active mood disorder diagnosis.
> - For hallucinations and inappropriate laughter, consider the client having schizophrenia.
> ---
> [Judge] Round 1 begins. Both sides should claim the diagnosis and reasons in this round.
>
> [Positive] I believe this client has a mood disorder.
> - The client's records show long-term mood fluctuations, ranging from depressive to elevated states, which is characteristic of bipolar disorder.
> - The client was previously hospitalized for a mood disorder and has been on mood-related medication, indicating a strong link to mood disorder symptoms.
> - Although clinician-rated assessments did not indicate significant depression or anxiety, the client's self-report showed moderate depression, which may better reflect their subjective emotional state.
>
> [Negative] I don't think this client has a mood disorder.
> - Clinician-rated assessments show no signs of depression, anxiety, or suicidal tendencies—key indicators of mood disorders.
> - Although the client initially improved with mood-related medication, new symptoms appeared despite continued medication use, suggesting a possible different mental condition.
> - Similar cases have inconsistent diagnoses, so relying solely on past hospitalization records is insufficient for a conclusive diagnosis.
>
> [Judge] Rebuttals missing. Continue.
>
> ---
>
> [Judge] Round 2 begins, both sides should refute the opposite side.
>
> [Positive]
> - Self-reported symptoms indicate depression and suicidality, with a PHQ-9 score of 19 and daily suicidal ideation. These are core indicators of mood disorders, even if clinician-rated scales show minimal symptoms.
> - New symptoms during continuous medication do not rule out mood disorders. Mood disorders often present episodically or with partial response to treatment.
> - Historical diagnosis and treatment response support a mood disorder. The client was previously hospitalized for an affective disorder and showed improvement with mood stabilizers.
>
> [Negative]
> - Mood fluctuations alone do not confirm a mood disorder. The observed shifts could be part of other psychiatric conditions such as borderline personality traits or stress-induced responses. Symptom variability requires a broader differential diagnosis.
> - Past hospitalization and medication use do not confirm the current diagnosis. Historical treatment for a mood-related condition is not definitive evidence of a present mood disorder, especially when recent clinician-rated assessments show no active depressive or anxious symptoms.
> - Self-report scores must be interpreted cautiously. While PHQ-9 indicates moderate depression, clinician-administered scales (HAMD, HAMA, BPRS) consistently show no depressive, anxious, or suicidal signs. Clinical observation remains the gold standard.
>
> [Judge] Certain points are not discussed. Continue.
>
> ---
>
> [Judge] Round 3. Please discuss the possibility of a performance conflict between self-report and clinician assessment, in one sentence.
>
> [Positive] The client's symptoms are suppressed by medicine.
>
> [Negative] The client may have schizophrenia.
>
> [Judge] All symptoms discussed. Debate ends.
> The client doesn't have a mood disorder. Although the client was previously hospitalized and prescribed medication related to mood disorders, multiple clinician-administered scales did not indicate typical symptoms of depression or anxiety. Moreover, the client's current symptoms are not alleviated by mood disorder medications, raising the possibility of other psychiatric conditions with overlapping features, such as schizophrenia.
>
> **Q2.** We designed the prompt of the Judge Agent to simulate a debate format, with a minimum of 2 rounds and a maximum of 4. At the beginning of each round, the prompt guides the Pro and Con agents through a structured process: Round 1 for opening statements, Round 2 for rebuttals, Round 3 for clarifying unresolved points, and the final round for closing arguments. After each round, the Judge Agent evaluates whether the debate should continue, provides reasoning, and reflects on the arguments. At the end of the debate, the judge delivers a diagnosis with reasons. The example in Q1 demonstrates a case where the debate concluded in three rounds.
>
> **Q3.** Thanks for your suggestion. We are testing on changing the temperature of the angel and the different prompt order of the debate part. We will display the result as soon as we complete the experiment.
>
> **Q4.** The debate agents generate reasons for their position by points. We evaluated their performances by three licensed psychiatry experts on our co-author clinician’s team. Metrics are clinical coherence (0-2 points),  information completeness (0-2 points), and clarity of expression (0-2 points).
>
> Clinical Coherence: whether reasoning aligns with clinical logic and evidence.
> - 2 = Fully coherent and professionally sound
> - 1 = Minor logical flaws
> - 0 = Major errors or conflicts
>
> Information Completeness: inclusion of key diagnostic elements (e.g., symptoms, scales, course, exclusions, impairment).
> - 2 = Comprehensive and sufficient
> - 1 = Missing important elements
> - 0 = Lacks key diagnostic info
>
> Clarity and Expression: clarity and professionalism of the explanation.
> - 2 = Clear and professionally worded
> - 1 = Some ambiguity or terminology misuse
> - 0 = Unclear or confusing
>
> Here are the assessment results:
> |Model|coh1|com1|clarity1|coh2|com2|clarity2|coh3|com3|clarity3|avg_coh|avg_com|avg_clarity|
> |:-------|:-----:|-----:|-----:|-----:|-----:|-----:|-----:|-----:|-----:|-----:|-----:|-----:|
> GPT-4o|0.5|0.9|0.7|0.2|0.5|0.9|0.6|1|1.1|0.43|0.5|0.9|
> Angel.R|0.9|1.1|0.9|0.6|0.7|1.2|1|1.3|1.3|0.87|1.03|1.17|
> Angel.D|0.8|1.2|0.9|0.6|0.6|1|0.9|1.3|1.2|0.83|1.03|0.73|
> Angel.C|0.9|1.2|1|0.7|0.8|1.1|1.1|1.4|1.3|1.27|1.4|1.13|
> multi-Angels|1.1|1.3|1.1|0.8|0.8|1.1|1.1|1.1|1.1|1.3|1.3|1.1|
>
> We report Fleiss' Kappa as the measure of inter-annotator agreement (IAA).
> |Model|coherence|completeness|clarity|
> |:-------|:-----:|-----:|-----:|
> GPT-4o|0.6|0.26|0.13|
> Angel.R|0.55|0.37|0.19|
> Angel.D|0.69|0.28|0.25|
> Angel.C|0.69|0.33|0.27|
> multi-Angels|0.67|0.45|0.54|
>
> The results demonstrate the high quality of our generated outputs. **The higher inter-annotator agreement suggests that the outputs of our multi-agent system are more consistent, clinically aligned, and better recognized by experts.**
>
>  **Q5.** Thank you sincerely for the thoughtful suggestion. Our judge is entirely language-model-based and is designed to simulate a debate adjudicator rather than proposing new diagnostic hypotheses. The DSM-5 alignment is primarily handled in earlier stages by the three independent diagnostic agents, whose outputs serve as the “debate material.” In this setting, the Judge is more like a referee than a debater, who does not gather evidence or match symptoms directly to criteria.

---

> > ### Author Response · Authors · 2025-08-03
> > **Q3. Stability of framework's output**
> >
> > In this ablation study, we evaluate the robustness of our framework under two settings:
> >
> > 1. Input Order Sensitivity:
> > We use the random.shuffle() function to randomize the input order during the debate stage:
> > Setting|ACC|MCC|
> > |:----:|----:|----:|
> > multi-Angels (ordered input)|0.925|0.834
> > multi-Angels (shuffled input)|0.927|0.838
> >
> > 2. Temperature Sensitivity:
> > We vary the temperature of GPT-4o to assess the output stability of Angel.R:
> > Setting|ACC|MCC|
> > |:----:|----:|----:|
> > Angel.R$_{temp=1}$(default)|0.920|0.829|
> > Angel.R$_{temp=0.7}$|0.922|0.836|
> > Angel.R$_{temp=0.3}$|0.923|0.836|
> > Angel.R$_{temp=0.0}$|0.911|0.808|
> >
> > These results indicate that variations in input order and temperature have minimal impact on diagnostic performance, demonstrating the stability and robustness of our framework.

---

> > > ### Comment · Reviewer_HsoZ · 2025-08-06
> > >
> > > The authors have provided examples and partially answered my previous questions. However, "We designed the prompt of the Judge Agent to simulate a debate format, with a minimum of 2 rounds and a maximum of 4. " does this method have any justification? Why a debate round number between 2 and 4 is optimal?

---

> > > > ### Author Response · Authors · 2025-08-07
> > > > **Design Rationale for the 2–4 Round Debate Structure**
> > > >
> > > > Thank you for your thoughtful question. Our debate stage is intentionally designed to simulate the classical four-stage structure commonly used in formal debates:
> > > >
> > > > Round 1 – Opening Statements: Both Pro and Con agents present their initial diagnostic positions based on the available information.
> > > >
> > > > Round 2 – Rebuttals: Each agent critically responds to the opponent's arguments, defending their own stance and challenging the other side’s reasoning.
> > > >
> > > > Round 3 – Free Debate (Optional): If the Judge Agent identifies unresolved or ambiguous points after the first two rounds, it initiates a more open-ended exchange to further clarify conflicting perspectives.
> > > >
> > > > Round 4 – Closing Statements (Optional): If the discussion remains inconclusive or highly contentious, both agents are given one final opportunity to summarize their most compelling arguments in a concise manner.
> > > >
> > > > This flexible structure allows the system to adapt the level of deliberation to the complexity of the case, balancing efficiency with thoroughness in clinical decision-making. The minimum of two rounds ensures that both sides have the opportunity to present and respond to each other's views—this establishes a meaningful diagnostic exchange. If the Judge Agent determines that all relevant points have been sufficiently addressed by the end of Round 2, the debate can be concluded early. However, when the case involves ambiguity or conflicting symptom interpretations, the Judge may trigger additional rounds to promote deeper reasoning.
> > > >
> > > > In practice, we found that approximately 80% of cases concluded after just two rounds of debate, as the key diagnostic arguments had already been clearly presented and addressed. The remaining cases required only one additional round, with all debates concluding within three rounds. This observation supports the efficiency of our debate framework in ensuring thorough reasoning when needed, while avoiding unnecessary verbosity or repetition.

---

> ### Comment · Area_Chair_weSh · 2025-08-02
> **Please read author rebuttal**
>
> Dear Reviewer,
>
> The author has provided a rebuttal to respond to your comments. Please have a read on the author response and discuss with author if necessary.
>
> Thanks,
>
> AC Team

---

### Note · Authors · 2025-08-12

We sincerely thank the reviewers for their thoughtful engagement and kind recognition of our work. Through the discussion process, we have addressed all major concerns raised and will incorporate key clarifications, additional experiments, and methodological details into the revised manuscript to further strengthen its quality. Below is a summary of the points discussed with reviewers:

In our rebuttal, we provided additional clarifications on the **motivation and technical contributions** of our diagnostic framework, which were acknowledged by Reviewer EzJT. Building on this, we further elaborated on **the design and role of key framework components**. Specifically, at the request of Reviewer HsoZ and Reviewer jxzA, we supplemented details about the debate mechanism and its functional role, supported by concrete examples. Additionally, for Reviewer 6o25, we strengthened the **theoretical basis of our Item-Level Analysis** with relevant literature references.

Regarding experimental validation, we addressed several key concerns raised by the reviewers. First, at the suggestion of Ethics Reviewer VGQ5, we analyzed **subgroup accuracy** by age and gender, confirming no significant bias in our framework. We also evaluated **medical LLMs as backbone or baselines** (Reviewers EzJT and 6o25), highlighting their current limitations. To further substantiate our methodology, we conducted **statistical significance testing and ablation studies on retrieval components** (Reviewer jxzA), as well as human evaluations on diagnostic reasoning (Reviewers HsoZ and 6o25), both of which affirmed the high quality and reliability of our approach.

Finally, concerning the **evaluation of our synthetic dataset**, we note that the original submission already included comprehensive automated assessments covering statistical density, data quality, machine learning efficacy, privacy preservation, and logistic detectability. In response to Ethics Reviewer VGQ5’s suggestion, we also conducted a clinician audit, which verified that our synthetic data exhibits strong plausibility and diagnostic value comparable to real clinical cases.

We appreciate the constructive feedback from all reviewers, which has been invaluable in refining both the technical rigor and clarity of our work.

---

### Decision · Program_Chairs · 2025-09-17

**Decision:**

Accept (poster)

**Comment:**

**Summary**
This paper presents a novel multi-agent framework, MoodAngels, designed for psychiatric diagnosis of mood disorders, and introduces a synthetic dataset, MoodSyn, to address data privacy challenges.

**Strengths**
* This work introduces a novel synthetic dataset, MoodSync, comprising 25 clinical features—including 16 diagnostic questions, 8 standardized scale scores, and expert-verified labels—that provides clinically valid and privacy-preserving cases for mood disorder analysis, enabling robust AI development in computational psychiatry.
* The paper also presents a novel framework addressing critical challenges in AI-assisted psychiatry, such as symptom overlap and interpretability. Specifically, the paper authors develop a tailored, clinically-grounded multi-agent system (e.g., Angel.R, Angel.D, Angel.C) that integrates granular-scale analysis, DSM-5-informed retrieval, and structured debate mechanism.

**Weaknesses**
* The empirical performance gain from the complex debate mechanism over the simpler (but highly effective) Angel.R or a majority vote is modest, though its value for interpretability is recognized.
* The task is framed as a binary classification (mood disorder present/absent) rather than fine-grained differential diagnosis between specific disorders, which is a necessary scope limitation but reduces immediate clinical application.


**Reason for Accept/Reject**

*Borderline Accept* - The paper presents two novel and methodologically sound contributions—a privacy-preserving synthetic dataset and an interpretable multi-agent framework—that meaningfully advance the field of AI in psychiatry. The authors also successfully address key reviewer concerns during rebuttal by demonstrating ethical rigor and providing additional analyses, though the practical clinical impact may be currently limited by the binary classification task.

**Rebuttal Discussion**
Initial reviews were mixed, but through the rebuttal phase of auxiliary experimental results and clarification to concerns, the authors addressed the majority of raised issues, and this resulted in three reviewers (HsoZ, EzJT, jxzA) updating their scores positively, leading to a consensus for acceptance.

In address of ethical concern consideration, the paper includes a clinician audit of synthetic data and a detailed subgroup performance analysis (by age and gender) that confirmed no significant bias, as requested by the ethics reviewer.